# The cell adhesion molecule Sdk1 shapes assembly of a retinal circuit that detects localized edges

Pierre-Luc Rochon, Catherine Theriault, Aline Giselle Rangel Olguin, Arjun Krishnaswamy*

Department of Physiology, McGill University, Montreal, Canada

**Abstract** Nearly 50 different mouse retinal ganglion cell (RGC) types sample the visual scene for distinct features. RGC feature selectivity arises from their synapses with a specific subset of amacrine (AC) and bipolar cell (BC) types, but how RGC dendrites arborize and collect input from these specific subsets remains poorly understood. Here we examine the hypothesis that RGCs employ molecular recognition systems to meet this challenge. By combining calcium imaging and type-specific histological stains, we define a family of circuits that express the recognition molecule Sidekick-1 (Sdk1), which include a novel RGC type (S1-RGC) that responds to local edges. Genetic and physiological studies revealed that Sdk1 loss selectively disrupts S1-RGC visual responses, which result from a loss of excitatory and inhibitory inputs and selective dendritic deficits on this neuron. We conclude that Sdk1 shapes dendrite growth and wiring to help S1-RGCs become feature selective.

*For correspondence:
arjun.krishnaswamy@mcgill.ca

Competing interest: The authors declare that no competing interests exist.

## Introduction

In the retina, each of the ~46 types of retinal ganglion cell (RGC) synapses with a unique subset of amacrine (AC) and bipolar cell (BC) types to create circuits that detect a unique aspect of the visual scene (*Clark and Demb, 2016*; *Gollisch and Meister, 2010*; *Jadzinsky and Baccus, 2013*; *Sanes and Masland, 2015*; *Shekhar et al., 2016*; *Tran et al., 2019*; *Yan et al., 2020*). A growing body of work suggests that recognition molecules guide the neurites of newborn retinal neurons to grow into sublayers of a specialized neuropil, the inner plexiform layer (IPL), where they contact hundreds of potential synaptic targets (*Rangel Olguin et al., 2020*; *Sanes and Yamagata, 2009*; *Sanes and Zipursky, 2020*; *Zhang et al., 2017*).

The factors that guide developing arbors to synapse with appropriate targets within a layer are not well understood. An initial idea, called Peter's principle, posited that developing neurons synapse with nearby cells according to how often they make contact (*Binzegger et al., 2004*; *Peters and Feldman, 1976*; *Shepherd et al., 2005*). However, recent connectomic studies of the IPL demonstrate no obvious relationship between contact frequency and synapse number (*Briggman et al., 2011*; *Helmstaedter et al., 2013*). Instead, these connectomic data support a model in which neurons recognize targets in their immediate vicinity and synapse specifically with them. Key molecules in this recognition process are thought to be members of the immunoglobulin superfamily (IgSFs).

Briefly, IgSFs are adhesion molecules that bind to themselves (homophilic) or compatible IgSFs (heterophilic) across cell-cell junctions. It has been proposed that selective IgSF expression within synaptic partners allows their neurites to adhere and synapse when they encounter each other (*Sanes and Zipursky, 2020*). A recent study in mouse retina provides support for this view (*Krishnaswamy et al., 2015*). In this study, the IgSF Sidekick-2 (Sdk2) in VG3-ACs and W3B RGCs drives these neurons to connect with each other far more than they connect with the Sdk2-negative neurons they contact. Loss of Sdk2 ablates the enhanced VG3-W3B connectivity but does not alter the gross structure or

overlap of their arbors, suggesting that IgSFs increase the probability of synapses between this pair (*Krishnaswamy et al., 2015*). On the other hand, direction-selective RGCs require the IgSF contactin 5 (Cntn5) to grow dendritic branches in IPL layers bearing axons of their AC/BC partners. Loss of Cntn5 from these RGCs decreases their dendritic branches and reduces synaptic input (*Peng et al., 2017*), suggesting that IgSFs influence connectivity by regulating intralaminar dendritic growth. These studies suggest a common role for these IgSFs to stabilize/promote the growth of small dendrites that lead to synapses or suggest differing roles for IgSFs in synaptic specificity and neurite growth. Too few IgSFs have been studied in the context of mammalian circuit assembly to draw a firm conclusion. To gain more insight, we investigated the closest IgSF relative of Cntn5 and Sdk2, called Sidekick-1 (Sdk1), in retinal circuit assembly.

We show that Sdk1 is expressed by a family of five RGCs whose dendrites target IPL layers bearing the processes of five Sdk1-expressing interneurons (ACs and BCs). We uncover molecular markers for each Sdk1 RGC and applied these markers following calcium imaging to investigate their visual responses, discovering that the Sdk1 family includes two ON-direction-selective RGCs and an RGC (S1-RGC) sensitive to local edges. Loss of Sdk1 disrupted responses to visual stimuli on S1-RGCs without affecting other members of the Sdk1-RGC family. Using electrophysiological recordings, we show that Sdk1 loss reduced excitatory and inhibitory synaptic inputs to S1-RGCs, decreasing its firing to visual stimuli. These synaptic deficits were specific to S1-RGCs as stimulus-evoked responses on Sdk1+ ON-alpha RGCs were unaffected by Sdk1 loss. Finally, we show that the loss of Sdk1 does not alter the IPL layer targeting of S1-RGC dendrites but selectively reduces their intralaminar complexity and size. From our results, we conclude that Sdk1 is required for S1-RGCs to develop dendritic arbors and receive AC and BC synapses.

## Results

### Sdk1 labels a family of retinal circuits

To investigate the expression of Sdk1 across the retina, we used mice in which the *Sdk1* gene is disrupted by the presence of cDNA encoding a Cre-GFP fusion protein (*Sdk1*<sup>CG</sup>); as heterozygotes these lines allow selective access to Sdk1 neurons, as homozygotes they are Sdk1 nulls (*Krishnaswamy et al., 2015*; *Yamagata and Sanes, 2018*). Cross-sections and whole mounts from these animals showed GFP+ nuclei within subsets of neurons expressing the BC marker Chx10, the AC marker Ap2-α, and the RGC marker RBPMS (*Figure 1A–F*); no labeling was found in horizontal cells and photoreceptors. Counting double-labeled cells showed ~7795 Sdk1-BCs//mm², ~350 Sdk1-ACs /mm², and ~390/mm² Sdk1-RGCs (*Figure 1G*), suggesting that Sdk1 labels several retinal circuits. To address this, we set out to match GFP+ neurons to retinal subtypes.

### Sdk1 defines a family of five RGCs

Publicly available single RGC sequencing atlases indicate that Sdk1 is highly expressed in approximately six RGC clusters (*Rheaume et al., 2018*; *Tran et al., 2019*). Two clusters correspond to alpha RGCs with sustained responses to light onset (ONα-RGCs) and type 2 melanopsin-positive RGCs (M2-RGCs). Four others predicted novel RGCs types (*Figure 1—figure supplement 1A*). To map these clusters onto GFP+ RGCs, we compared Sdk1 clusters to all RGCs and identified five genes that combinatorially label each Sdk1-RGC type: (1) Onα-RGCs should strongly express neurofilament heavy chain isoform (Nefh), osteopontin (Ost), and the intracellular calcium buffer calbindin (Calb); (2) M2-RGCs should express high levels of Ost+, low levels of Nefh, and low levels of Calb; (3) two predicted novel RGCs should express the steroid hormone receptor Nr2f2 (Nr2f2), (4) with one of these also expressing Calb; (5) another novel RGC should express the Pou-domain containing transcription factor (Brn3c); and (6) a final RGC should not express any of these genes (*Figure 1—figure supplement 1B*).

We took advantage of the mosaic arrangement of retinal types as an initial test of these predictions. Briefly, retinal neurons of the same type are often spaced apart at a characteristic distance, whereas neurons of different types are often spaced randomly (*Keeley et al., 2020*; *Sanes and Masland, 2015*). The degree of regularity varies, but for many types, the density of cells labeled with a candidate marker will drop at short distances from a reference cell if the candidate labels a single type (*Keeley et al., 2020*; *Sanes and Masland, 2015*). Sdk1<sup>CG</sup> whole mounts stained with antibodies

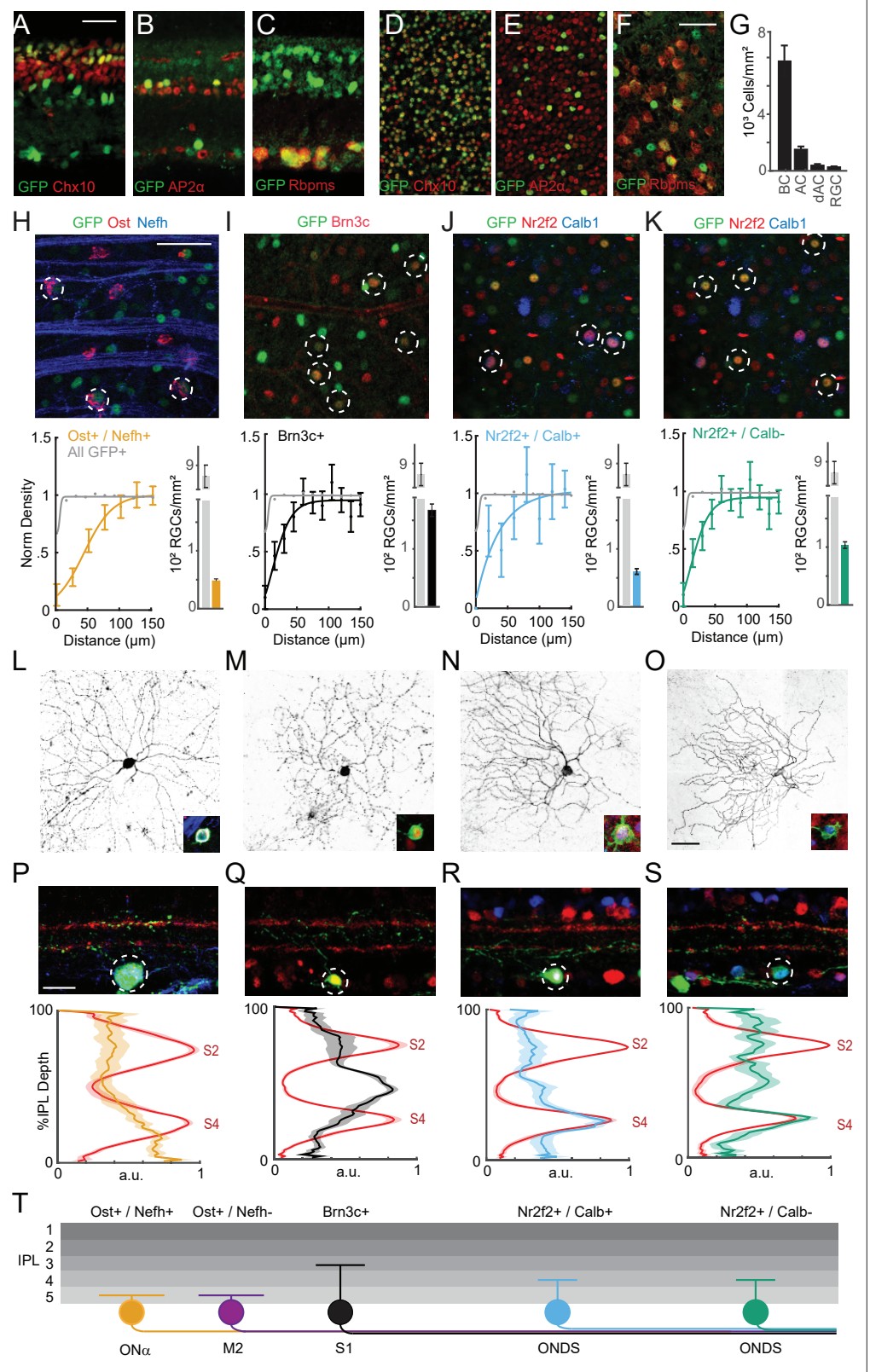

**Figure 1.** Sidekick-1 (Sdk1) labels a family of retinal ganglion cell (RGC) types. (**A–F**) Sample cross-sections (**A–C**) and whole mounts (**D–F**) from Sdk1CG mice stained with antibodies to GFP and the bipolar cell (BC) marker Chx10 (**A, D**), amacrine (AC) marker AP2-α (**B,E**), and RGC marker RBPMS (**C, F**). Scale = 25 μm. (**G**) Bar graph showing density of BCs, ACs, displaced amacrine cells (dACs), and RGCs expressing Sdk1 computed from experiments

*Figure 1 continued on next page*

*Figure 1 continued*

like those shown in (**D–F**) (n = 14 fields from three animals for each experiment). (**H–K**) Top: GFP-stained sample Sdk1^CG retinal whole mounts co-stained with antibodies to osteopontin (Ost) and neurofilament heavy chain (Nefh) (**H**), Brn3c (**I**), or Nr2f2 and calbindin (Calb) (**J**, **K**). Dotted circles indicate co-labeled neurons and scale = 50 μm. Bottom: normalized density recovery profiles and average density of co-labeled RGCs measured from corresponding experiments shown in the top row. Colored traces indicate density recovery profiles for co-labeled neurons; gray traces indicate density recovery profiles for all GFP+ cells in the GCL (n = 13–18 fields from six animals for each experiment). (**L–O**) Images showing the dendritic morphology of individually labeled Ost+/Nefh+ RGCs (**L**), Brn3c+ RGCs (**M**), Nr2f2+/Calb+ RGCs (**N**), and Nr2f2+/Calb- RGCs (**O**) in *Sdk1*^CG/+ retinal whole mounts. Inset shows marker expression in the soma. (**P–S**) Top: images showing the laminar morphology of individually labeled Ost+/Nefh+ RGCs (**P**), Brn3c+ RGCs (**Q**), Nr2f2+/Calb+ RGCs (**R**), and Nr2f2+/Calb- RGCs (**S**) in *Sdk1*^CG/+ retinal cross-sections. Red staining indicates VAcht, a marker for sublaminae 2 (S2) and 4 (S4) (scale = 50 μm). Bottom: inner plexiform layer (IPL) linescans measured from corresponding experiments shown in the top row. Red traces show VAChT intensity, and colored traces show reporter intensity measured from experiments like those shown in the top row (n = 6–12 fields per RGC type from more than 16 animals). (**T**) Summary cartoon of the Sdk1 RGC family showing Ost+/Nefh+ Onα-RGCs, Brn3c+ S1 RGCs, Ost+/Nefh- M2-RGCs, and Nr2f2+/Calb+ and Calb- ON-DSGCs.

The online version of this article includes the following figure supplement(s) for figure 1:

**Figure supplement 1.** Molecular taxonomy of Sidekick-1-retinal ganglion cells (Sdk1-RGCs).

**Figure supplement 2.** Marker gene histology for Sidekick-1-retinal ganglion cells (Sdk-RGCs).

**Figure supplement 3.** Sidekick-1-retinal ganglion cell (Sdk1-RGC) projections to retinorecipient areas.

**Figure supplement 4.** Sidekick-1 (Sdk1) labels five kinds of interneurons.

to Nefh and Ost showed a pair of Nefh+/Ost+ and Nefh-/Ost+ mosaics spaced at distances expected for the ONα-RGC (*Figure 1H*, *Figure 1—figure supplement 2A*) and M2-RGC (*Figure 1—figure supplement 1C*), respectively. A high-density mosaic was labeled by Brn3c (*Figure 1I*, *Figure 1—figure supplement 2C*), and a final pair of mosaics was found to be Nr2f2+/Calb- and Nr2f2+/Calb1+ (*Figure 1J and K*, *Figure 1—figure supplement 2B*). We were unable to find an RGC that corresponded to the sixth Sdk1 cluster, as a stain with all these RGC markers and Ap2-α labeled all GFP+ neurons in the GCL (*Figure 1—figure supplement 1E–G*). Thus, we provide molecular definition for five Sdk1-RGCs, which include three predicted novel types.

We used two approaches to define the anatomy of Sdk1-RGCs. In one approach, retrogradely infecting AAVs bearing Cre-dependent reporters were delivered to the lateral geniculate nucleus (LGN) or superior colliculus (SC) of Sdk1^CG mice. In the other, tamoxifen was used to drive reporter expression in a related strain (Sdk1^CreER) whose *Sdk1* gene is disrupted with cDNA encoding a Cre-human estrogen receptor (CreER) fusion protein (*Krishnaswamy et al., 2015*). Staining retinae from these mice with our panel of molecular markers revealed the morphology of two Sdk1-RGCs: (1) Ost+/Nefh+ RGCs bore large somas and wide dendritic arbors confined to IPL sublamina 5 and matched previous descriptions of ONα-RGCs (*Figure 1L and P*; *Krieger et al., 2017*; *Krishnaswamy et al., 2015*); and (2) Brn3c+ RGCs had small somas and grew dendritic arbors confined to the center of sublamina 3 (*Figure 1M and Q*). We refer to these Brn3c+ neurons as S1-RGCs. Ost+/Nefh- and Nr2f2+ RGCs were rarely observed using this method. These results suggest that only a subset of Sdk1-RGCs project strongly to the LGN and SC (*Figure 1—figure supplement 3A and D*).

One possibility for our relatively poor labeling of Ost+/Nefh- and Nr2f2+ RGCs is that these RGCs innervate 'non-imaging-forming' brain regions more strongly than they innervate the LGN and SC (*Martersteck et al., 2017*). Consistent with this idea, brain sections taken from intraocularly infected Sdk1^CG mice showed reporter-labeled RGC axons in the non-image-forming medial terminal nucleus (MTN) and olivary pretectal nuclei (OPN; *Figure 1—figure supplement 3B and C*). To sparsely label these RGCs, we tamoxifen-treated *Sdk1*^CreER/+ mice crossed to Cre-dependent reporters and stained their retinae with Ost, Nefh, Nr2f2, and Calb. These experiments revealed that both Nr2f2+/Calb+ and Nr2f2+/Calb- RGCs grew oblong dendritic arbors confined to IPL sublamina 4 (*Figure 1N, R, O and S*). We observed a few Ost+/Nefh- RGC somas in fields labeled densely by reporter (*Figure 1—figure supplement 1H*), but we were unable to mark them individually, likely due to their low density (~20/mm$^2$) (*Sanes and Masland, 2015*). Thus, Sdk1 labels a family of five RGCs, which include image- and non-image-forming types.

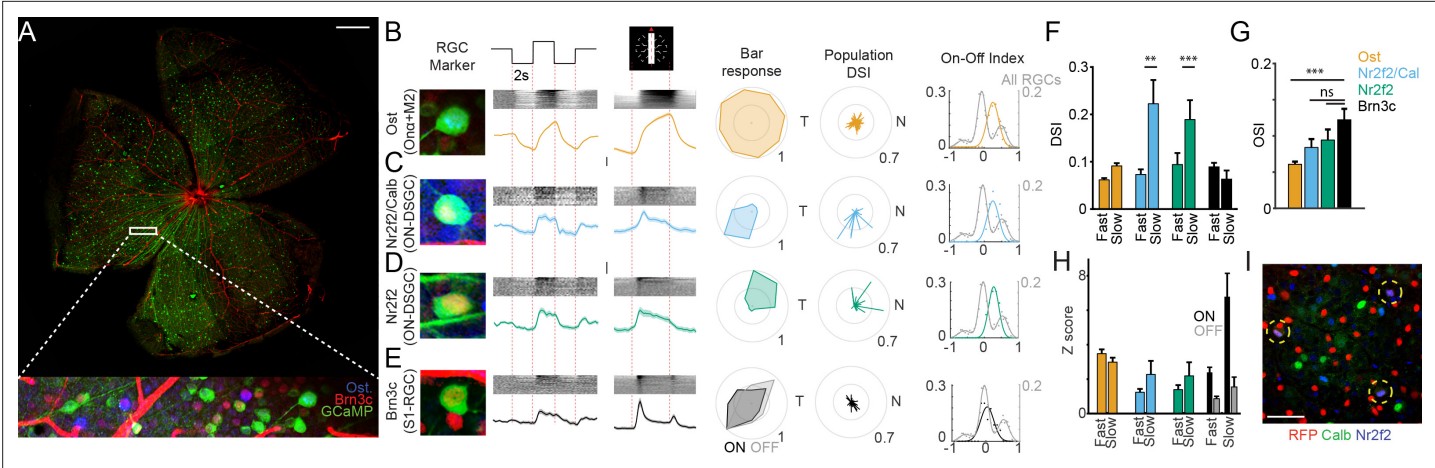

**Figure 2.** Function of Sidekick-1 (Sdk1) circuits. (**A**) Sample whole mount from a Sdk1$^{CG}$ retina infected with AAV-GCaMP6f labeled with the positions of a typical two-photon field (scale = 500 μm). Inset magnifies the boxed field after immunostaining and shows GCaMP-labeled Sdk1-RGCs (green) stained with Brn3c (red) and osteopontin (Ost, blue). (**B–E**) Magnified somata image, average full-field response, average bar response, sample polar plot, population direction-selective indices (DSI), and ON-OFF index for Ost+ (**B**, n = 107 cells from 18 retinae), Nr2f2+/Calb+ (**C**, n = 41 cells from 6 retinae), Nr2f2+/Calb- (**D**, n = 34 cells from 6 retinae), and Brn3c+ RGCs (**E**, n = 95 cells from 6 retinae). Raster above each averaged trace shows every response within each retinal ganglion cell (RGC) group. Each RGC shows a characteristic pattern of responses to stimuli. ON-OFF index distribution from a mouse line in which all RGCs express GCaMP6f (all RGCs; n = 1426 cells from five retinae) is shown for comparison. Vertical scales = z-score of 2. (**F**) Average DSI computed from bar stimuli moving at ~1000 μm/s (fast) or ~200 mm/s (slow) for each Sdk1-RGC group. Significance: **p<0.01; ***p<0.001. (**G**) Average orientation-selective indices computed from bar stimuli for each Sdk1-RGC group. (**H**) Average response magnitude to bars moving at ~1000 μm/s (fast) or ~200 mm/s (slow) for each Sdk1-RGC group. (**I**) Whole-mount retina from the Pcdh9-Cre line crossed to Cre-dependent reporter stained for Nr2f2 and Calb. Triple-labeled cells are encircled (scale = 50 μm).

The online version of this article includes the following figure supplement(s) for figure 2:

**Figure supplement 1.** Registration of stained and two-photon imaged retinal fields.

**Figure supplement 2.** Molecular markers define retinal ganglion cell (RGC) response clusters.

## Sdk1 defines at least five interneuron types

We next asked if Sdk1-ACs and BCs target the same layers as Sdk1-RGCs. Staining reporter-labeled retinae from Sdk1$^{CreER/+}$ mice with Ap2-α revealed three kinds of Sdk1-ACs (*Figure 1—figure supplement 4A–C*). One of these had very large (>1 mm) diameter dendritic arbors located in sublamina 5 (*Figure 1—figure supplement 4A*), another had a similarly large arbor located in sublamina 3 and resembled descriptions of type 2 catacholaminergic ACs (*Figure 1—figure supplement 4B*; *Knop et al., 2014*; *Krishnaswamy et al., 2015*). A third Sdk1-AC grew comparatively narrower dendritic arbors that exhibited the characteristic waterfall appearance of the A17 AC (*Figure 1—figure supplement 4C*; *Masland, 2012*). Cross-sections from the same experiments also showed the presence of many rod BCs as judged by their anatomy (*Figure 1—figure supplement 4D*), as well as many BCs that targeted the interface between sublamina 3 and 4, matching reports of type 7 BCs (*Figure 1—figure supplement 4E*; *Wassle et al., 2009*). The IPL lamination profiles of Sdk1-ACs, BCs, and RGCs show that these cells target a common set of IPL laminae. Thus, we conclude that Sdk1 defines a family of neurons whose circuits are contained within the inner three layers of the IPL.

## Sdk1-RGCs include direction-selective and edges-detecting subtypes

Registering calcium imaged fields to post hoc stains assigns molecular identity to RGC response

To characterize the functional properties of the Sdk1 RGC family, we devised a procedure to relate marker-gene expression to neural response (*Figure 2A*, *Figure 2—figure supplement 1*). Briefly, retinal neurons in Sdk1$^{CG}$ mice were intraocularly infected with AAVs bearing Cre-dependent genetically encoded calcium indicators (GCaMP6f), and 2 weeks later, we imaged their responses to full-field flashes and bars moving at high or low velocity in eight different directions using two-photon microscopy. To register each two-photon field to the optic nerve head, we labeled blood vessels

fluorescently labeled with sulphorhodamine 101 and matched the vascular pattern (*Figure 2—figure supplement 1A and B*). Next, we fixed and stained retinae for GFP, Sdk1-RGC markers, and vessels, and reimaged these retinae with a confocal microscope (*Figure 2—figure supplement 1C*). Vascular patterns let us register confocal and two-photon fields (*Figure 2—figure supplement 1D–K*) and Ost distribution let us orient each retina along the dorsoventral and nasotemporal axes (*Bleckert et al., 2014*). Finally, regions-of-interest (ROIs) were drawn using the confocal image and applied to channels containing marker stains and to the two-photon images to extract responses (*Figure 2—figure supplement 1L–O*).

A balance between antibody species restrictions and throughput led us to group the low-abundance M2-RGC (<20/mm$^2$) with ONα-RGCs using the marker Ost. This registration procedure divided our GCaMP6f dataset into four marker-defined groups, with each bearing a characteristic response to visual stimuli and enrichment of RGC markers (*Figure 2*, *Figure 2—figure supplement 2A–D*). To check the consistency of these marker-defined groups, we compared the bar-evoked responses within each group to the average response of all four groups and asked if grouped traces were most similar to their group mean. To do this, we normalized each trace and computed its cosine similarity to the mean response of each group. Ost+ and Brn3c+ traces showed highest similarity to their own mean group responses. Nr2f2+ and Nr2f2+/Calb+ traces also showed high similarity to their mean group response but also showed similarity to each other's mean response, consistent with the strong resemblance in their visual responses (*Figure 2—figure supplement 2A–D*). These results indicate that our registration procedure can group molecularly and functionally similar RGCs.

## Ost+ RGCs show sustained responses to light onset

Ost+ RGCs (M2- and Onα- RGCs) showed sustained responses to the onset of a full-field flash and leading edge of the moving bar. Converting these bar responses to an ON-OFF index (see Materials and methods) and comparing the distribution of these indices to those computed from a dataset in which all RGCs express GCaMP6f (*Figure 2B–E*, ON-OFF Index All RGCs; Slc17a6-Cre::Ai95D) emphasize this observation (*Figure 2B*). Plotting bar direction versus evoked response on a polar plot showed no obvious preference for motion direction (*Figure 2B*). Converting each polar plot to a direction-selective index (DSI) and preferred angle and viewing the entire Ost+ population on polar axes confirmed this picture, showing weak directional tuning with no systematic preference for stimuli moving along any cardinal axis (*Figure 2B*). Comparing the average DSI within this group for fast- or slow-moving bars was similar, indicating a lack of motion selectivity within the Ost+ RGC group (*Figure 2F*). These results are consistent with previous reports of ONα- and M2-RGCs (*Berson et al., 2010*; *Krieger et al., 2017*).

## Nr2f2+ RGCs are ON-direction selective

On the other hand, both Nr2f2+ RGC types gave transient responses to the leading edge of the moving bar that varied systematically with bar direction. Aligning these responses to retinal orientation revealed polar plots that pointed ventrally for Nr2f2+/Calb+ RGCs (*Figure 2C*) or moving dorsally for Nr2+/Calb- RGCs (*Figure 2D*). DSI polar plots for these two populations showed the same respective biases for ventral or dorsal motion and reducing bar velocity caused the average DSI of both RGC types to increase (*Figure 2F*), indicating that Nr2f2+ RGCs encode the direction of slow-moving bright stimuli. Taken together with our anatomical results, Nr2f2+ RGCs strongly resemble ON-direction-selective RGCs (ON-DSGCs), which grow dendrites in sublamina 4 and comprise three subtypes attuned to motion toward either dorsal, ventral, and nasal poles of the retina (*Dhande et al., 2013*; *Sanes and Masland, 2015*; *Yonehara et al., 2009*). Staining reporter-labeled retinae from the Pcdh9-Cre line, which marks ventrally tuned ON-DSGCs (*Lilley et al., 2019*; *Matsumoto and Yonehara, 2018*), showed overlap with ventral motion-selective Nr2f2+/Calb+ RGCs (~3% of reporter-labeled cells; 1035 Pcdh9-Cre cells/mm$^2$; *Figure 2I*). Thus, we conclude that Nr2f2+ RGCs likely correspond to a pair of ON-DSGCs.

## Brn3c+ S1-RGCs are ON-OFF RGCs that respond to bars traveling along the same axis

S1-RGCs (Brn3c+) showed ON responses to a full-field flash but responded to both the leading and trailing edge of the bright moving bar (*Figure 2E*). These results indicate that S1-RGCs can respond

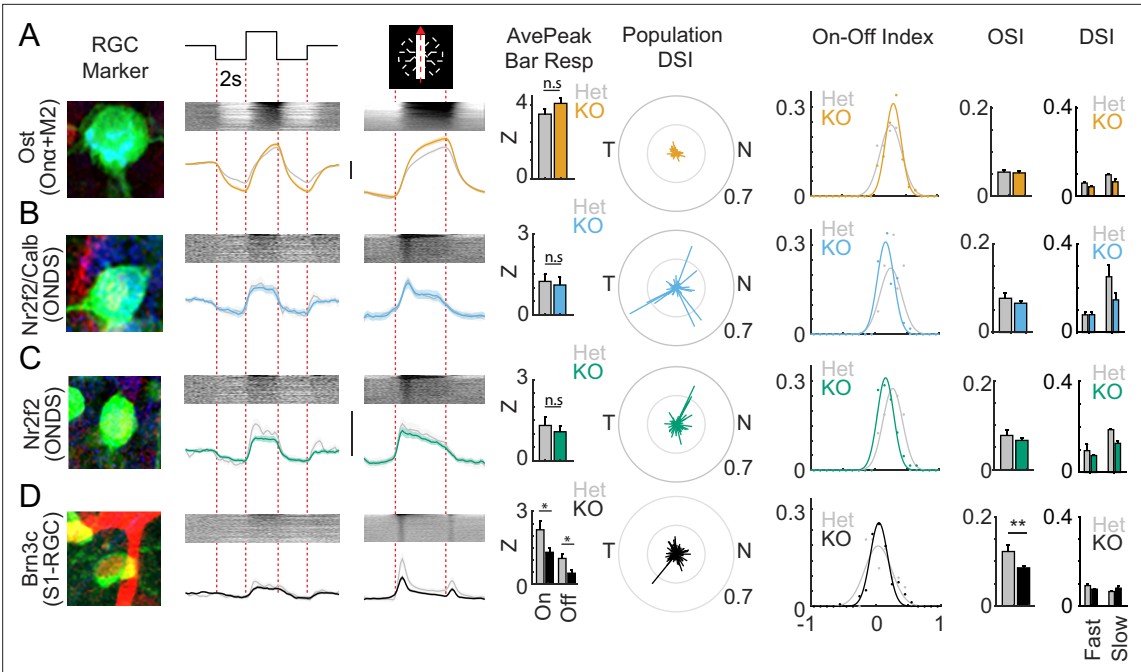

**Figure 3.** Sidekick-1 (Sdk1) loss causes selective deficits in S1-RGCs. (**A–D**) Magnified somata image, average full-field response, average bar response, average peak response to bars, population direction-selective indices (DSI), ON-OFF index, mean orientation-selective index (OSI), and mean DSI corresponding to Ost+ (**A**, n = 97 cells from 10 retinae), Nr2f2+/Calb+ (**B**, n = 59 cells from 6 retinae), Nr2f2+/Calb- (**C**, n = 77 cells from 6 retinae), and Brn3c+ RGCs (**D**, n = 147 cells from 7 retinae) in Sdk1null retinae. Grayed traces and bars show the same measurements from Sdk1 heterozygotes (Het) (vertical scales = z-score of 2). Sdk1 loss alters Brn3c retinal ganglion cell (RGC) visual responses. Significance: *p<0.05.

The online version of this article includes the following figure supplement(s) for figure 3:

**Figure supplement 1.** Expression of Sidekick-2 (Sdk2) in Sdk1-RGCs.

to both bright and dark stimuli and suggest that large full-field stimuli may recruit surround mechanisms that attenuate S1-RGC responses. Plotting S1-RGC bar responses against bar direction showed that many cells respond best to bars traveling slowly (200 µm/s) along the same axis, suggesting that these neurons detect stimulus orientation (*Figure 2E and H*). Computing an orientation selectivity index (OSI) for these cells showed a higher orientational preference in this neuron as compared with the other Sdk1-RGCs (*Figure 2G*); however, these values are significantly weaker than the OSI found in recently described orientation-selective RGCs (*Nath and Schwartz, 2016*; *Nath and Schwartz, 2017*). Thus, we conclude that S1-RGCs are ON-OFF cells that respond best to edges that fall within their receptive fields.

## Sdk1 loss selectively impairs S1-RGC visual responses

### Sdk1 loss impairs a subset of types in the Sdk1 RGC family

IgSFs like Sdk1 are thought to guide RGCs to synapse with specific AC and BC types to create feature-selective neural circuits (*Sanes and Zipursky, 2020*). Given that Sdk1 labels a family of five RGCs and interneurons whose processes overlap in a common set of lamina, we next asked if Sdk1 is important for the development of functional properties in Sdk1 RGCs. To test this idea, we labeled Sdk1 neurons with GCaMP using intraocular AAV injections in *Sdk1*^CG/CG mice, imaged their responses, and grouped these responses using type-specific markers. The responses of Sdk1-null Ost+ RGCs, which include Onα- and M2-RGCs, resembled their heterozygote counterparts (*Figure 3A*). Nr2f2+ RGCs, which include two kinds of ON-DSGCs, showed subtle changes in their average responses to full-field and moving bar stimuli (*Figure 3B and C*) with altered directional preference and selectivity, particularly in the NR2f2+/Calb+ group (*Figure 3B*). S1-RGCs were most affected by Sdk1 loss and showed a significant decrease in response magnitude to bar stimuli (*Figure 3D*). These results indicate that Sdk1 loss strongly impairs S1-RGCs. Interestingly, single-cell sequencing data showed high levels of Sdk2 in other Sdk1-RGCs relative to S1-RGCs, suggesting that there could be functional redundancy

in Onα-, M2-, and ON-DSGCs (*Figure 3—figure supplement 1A*). We partially confirmed this expression pattern by staining retinae from a Sdk2-CreER knock-in line crossed to reporters with Ost, Nefh, Nr2f2, and Calb antibodies and found many Onα- and Nr2f2+/Calb+ RGCs (*Figure 3—figure supplement 1B*). However, whole-retina quantitative PCR measurement of *Sdk2* transcript levels did not show a significant upregulation in Sdk1 nulls as compared to heterozygotes (*Figure 3—figure supplement 1C*), although an upward trend was detected. Based on these results, we focused on the S1-RGC for the rest of this study.

## Sdk1 loss selectively disrupts S1-RGC visual responses

Retrograde viral injections in the LGN/SC of *Sdk1*CG/+ offered us a way to study the role of Sdk1 in S1-RGCs and use Sdk1+/Sdk2+ ONα-RGCs as an internal comparison (*Figure 4A and B*). Recordings from *Sdk1*CG/+ mice showed many RGCs whose anatomical and functional properties matched one of these two groups. S1-RGCs were easily targeted for loose-patch recordings by their small soma and showed ON and OFF responses to a moving bar that were strongest for motion along a single axis as judged by their elongated profiles on a polar plot (*Figure 4A–C*). Similar recordings from nearby large-soma ONα-RGCs showed sustained responses to light onset with little tuning for motion direction (*Figure 4C*).

Our calcium imaging experiments showed that S1-RGCs have strong surround suppression. To relate these signals to spiking behavior, we recorded responses from control S1-RGCs to an expanding spot centered over their receptive field that flashed ON and OFF. S1-RGCs showed ON and OFF responses to this stimulus that were strongly suppressed by spot size, with OFF responses nearly silenced by spots exceeding ~300 μm (*Figure 4D, F, G*, *Figure 4—figure supplement 1*). Both ON and OFF components were most strongly activated by spot sizes close to the diameter of S1-RGC dendritic arbors, indicating that the receptive field center on these neurons is ~120–150 μm (*Figure 4F and G*). Recordings from Sdk1 heterozygote ONα-RGCs displayed higher baseline spiking than S1-RGCs and sustained ON responses that were poorly suppressed by large stimuli (*Figure 4E*), consistent with their full-field and moving bar responses in our calcium imaging experiments. Similar responses were found in wild-type Onα-RGCs, bearing two copies of Sdk1, labeled with the Kcng4-Cre line crossed to Cre-dependent reporters (*Figure 4—figure supplement 2A and B*). In contrast, the same recordings from S1-RGCs in *Sdk1*CG/CG retinae showed a dramatic loss of responsivity to dark stimuli and significantly weaker responses to ON stimuli (*Figure 4D, F, G*). Recordings from nearby Sdk1-null ONα-RGCs showed comparable responses to their control counterparts (*Figure 4E and H*), indicating that Sdk1 loss selectively affects S1-RGCs.

We found that most S1-RGCs showed responses to axial bar motion, similar to what we saw in our calcium imaging experiments. We suspected the inability to align bars with S1-RGC-receptive fields in the imaging studies could have activated their strong surround and attenuated their responses to this stimulus. We revisited the idea that these neurons might exhibit sensitivity to stimulus orientation, presenting stationary bars whose width matched the receptive field size of S1-RGCs and rotated through eight different orientations. As expected from their responses to moving bars, S1-RGCs responded preferentially to both bright and dark bars oriented along a single axis (*Figure 4I*), whereas Onα-RGCs respond to bright bars alone with little response variation to bar orientation (*Figure 4J*). The same stimulus evoked poor responses from S1-RGCs in *Sdk1*CG/CG retinae (*Figure 4I,K,L*), with significantly weaker responses to dark and bright bars, confirming our results using bright and dark expanding spot stimuli. The same recordings from nearby Sdk1-null Onα-RGCs showed similar responses to those in controls (*Figure 4J and M*). Computing OSI values for these cells and comparing the mean OSI for S1-RGCs and ONα-RGCs in controls and knockout retina showed a selective reduction of orientation selectivity for S1-RGCs in the absence of Sdk1 (*Figure 4N*). Thus, we conclude that Sdk1 is required for S1-RGCs to develop normal responses to visual stimuli.

## Sdk1 loss impairs excitatory and inhibitory synaptic inputs to S1-RGCs

The deficits we observed on S1-RGCs in *Sdk1*CG/CG retinae might result from a loss of excitatory inputs, a change in inhibitory inputs, or both. To investigate this idea, we recorded synaptic currents from S1-RGCs and ONα-RGCs in control and Sdk1 null retinae and compared their responses to visual stimuli. We began with expanding spots, isolating excitatory and inhibitory inputs by holding neurons at –60 mV and 0 mV, respectively (*Figure 5A and B*). S1-RGCs in controls showed inward (excitatory)

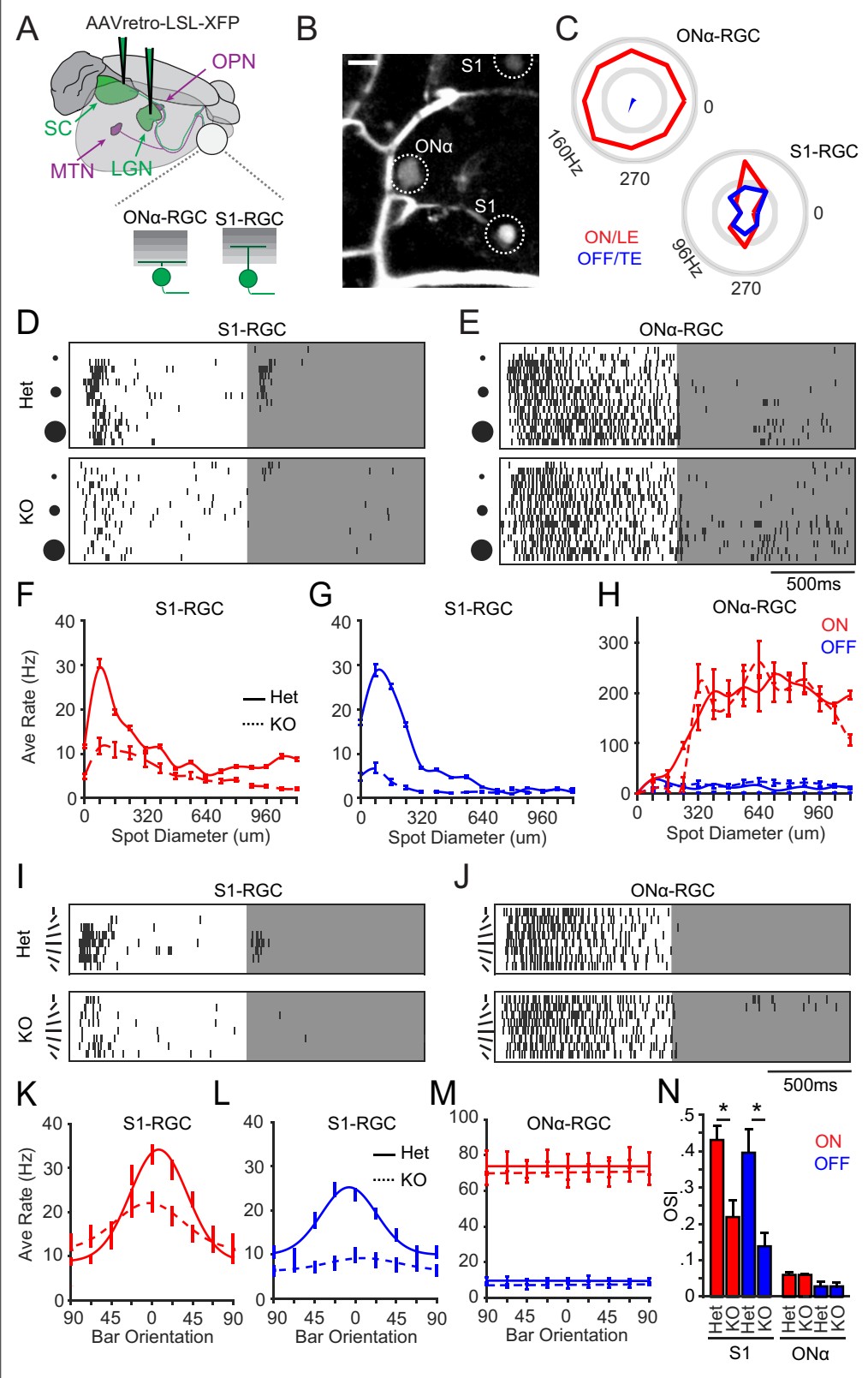

**Figure 4.** Selective loss of visual responses on S1-RGCs in the absence of Sidekick-1 (Sdk1). (**A**) Cartoon of S1-RGCs and ONα-RGCs labeled by delivering retrogradely infecting AAVs bearing Cre-dependent reporters in the lateral geniculate nucleus (LGN) or superior colliculus (SC) of Sdk1[CG] mice. Targets of other Sdk1-RGCs that project to olivary pretectal nuclei (OPN) and medial terminal nucleus (MTN) are also shown. (**B**) Sample two-photon

*Figure 4 continued on next page*

*Figure 4 continued*

image of a *Sdk1*^CG/+ retina labeled as described in (**A**) showing a large-soma ONα-RGC and a small-soma S1-RGC. Sulphorhodamine 101 labels vessels in the GCL. Scale = 25 µm. (**C**) Polar plots of spike responses to a bar moving in eight different directions recorded from example ONα- and S1-RGCs in experiments like those shown in (**B**). (**D, E**) Raster of spike responses to an expanding flashing spot recorded from example S1-RGCs (**D**) and Onα-RGCs (**E**) in *Sdk1*^CG/+ (Het) or *Sdk1*^CG/CG retinae (KO). (**F, G**) Average S1-RGCs firing rates versus bright (**F**) or dark (**G**) spot diameter measured from experiments like those shown in (**D**). (**H**) Average Onα-RGCs firing rate versus bright (ON) or dark (OFF) spot diameter measured from experiments like those shown in (**E**). (**I, J**) Raster of spike responses to centered dark or bright bar rotating through eight orientations recorded from S1-RGCs (**I**) and Onα-RGCs (**J**) in *Sdk1*^CG/+ (Het) or *Sdk1*^CG/CG retinae (KO). (**K, L**) Average firing rate versus bar orientation for S1-RGCs measured from experiments like those shown in (**I**). (**M**) Average firing rate versus bar orientation for ONα-RGCs measured from experiments like those shown in (**J**). (**N**) Average orientation selectivity indices computed for S1-RGC and ONα-RGC responses to the oriented bar stimulus in control (Het) and Sdk1-null (KO) retina (n = 7 for *Sdk1*^CG/+ ONα RGCs, n = 12 for *Sdk1*^CG/+ Brn3c RGCs, n = 14 for *Sdk1*^CG/CG Brn3c RGCs, n = 6 for *Sdk1*^CG/CG ONα RGCs; *p<0.05). RGC: retinal ganglion cell.

The online version of this article includes the following figure supplement(s) for figure 4:

**Figure supplement 1.** Visual responses of S1-RGCs to bright and dark stimuli.

**Figure supplement 2.** Physiology and anatomy of Sdk1^+/+ Onα-RGCs.

currents at –60 mV to both bright and dark spots that grew in strength until the spot reached ~150 µm in diameter and then steadily weakened (*Figure 5C and D*). Outward (inhibitory) currents at 0 mV were also found to both bright and dark spots but grew steadily with spot diameter (*Figure 5E and F*). Recordings from S1-RGCs in Sdk^CG/CG retinae showed significantly weaker inward and outward currents to bright and dark spots at all spot diameters (*Figure 5A–F*), indicating a loss of functional excitatory and inhibitory synapses on S1-RGCs in the absence of Sdk1. Whole-cell recordings from nearby ONα-RGCs showed inward and outward currents that were similar between controls and knockouts (*Figure 5—figure supplement 1A and B*), indicating that Sdk1 loss impairs synaptic inputs on S1-RGCs rather than generally affecting excitatory and inhibitory synaptic strength.

Finally, we examined the synaptic currents evoked on S1-RGCs to our oriented bar stimuli. Controls showed inward and outward currents to both bright and dark oriented bars, but the magnitudes of these currents varied with bar orientation (*Figure 5—figure supplement 2A and B*). Integrating these currents across the presentation time of each bar and normalizing the responses showed a systematic change in stimulus-evoked charge with bar angle. Orientations that produced the strongest excitation were orthogonal to those producing the strongest inhibition (*Figure 5—figure supplement 2E and F*). The same stimulus evoked significantly weaker inward and outward currents from S1-RGCs in *Sdk1*^CG/CG retinae, consistent with their reduced responses to the expanding spot stimulus (*Figure 5—figure supplement 2C, D,G–I*). Integrating these responses across stimulus presentation time and normalizing these responses to the average maximal response in controls showed that excitatory currents retained their preference for bar orientation, but the tuning of inhibitory inputs became less selective (*Figure 5—figure supplement 2G–I*). Thus, we conclude that Sdk1 is required for S1-RGCs to develop excitatory and inhibitory synaptic inputs.

## Sdk1 loss impairs S1-RGC dendritic development

Finally, we asked whether there are structural correlates of the reduced synaptic input to S1-RGCs. One possibility is that Sdk1-null S1-RGCs target inappropriate IPL sublayers and therefore cannot receive input from their interneuron partners. However, comparing the laminar position of S1-RGC dendrites in *Sdk1*^CG/+ or *Sdk1*^CG/CG retinal cross-sections showed no obvious difference between controls and nulls (*Figure 5G and H*). We next asked if Sdk1 loss impacts the lateral anatomy of S1-RGC dendrites. We found that the loss of Sdk1 led S1-RGCs to grow dendritic arbors that were less complex than their control counterparts (*Figure 5I*). Sdk1-null S1-RGC arbors contained similar numbers of dendritic branches (*Figure 5J*), but they were approximately half as long on average, which led to fewer intersections across the entire dendritic arbor as assessed by Sholl analysis (*Figure 5K–M*). Reduced S1-RGC dendritic arbor size did not alter the spatial distribution of these neurons (*Figure 5—figure supplement 1A and B*). These deficits arose with little gross change in the overall structure of the IPL, as assessed by staining with a variety of markers that label AC and BC

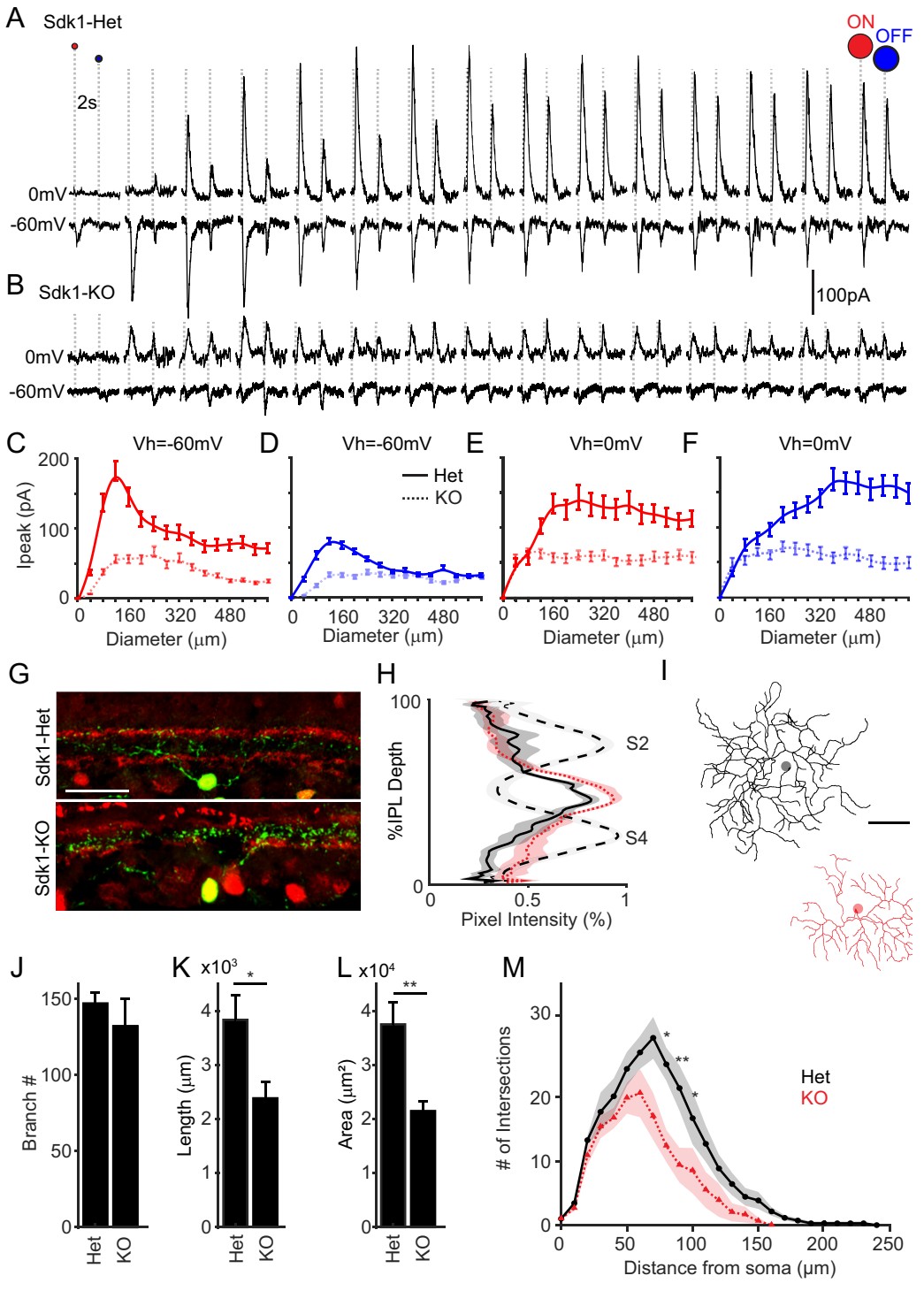

**Figure 5.** Sidekick-1 (Sdk1) loss causes S1-RGCs to lose synaptic inputs and dendritic arbor complexity. (**A–B**) Whole-cell recordings from S1-RGCs held at potentials to isolate excitatory (~–60 mV) or inhibitory (~0 mV) currents to an expanding flashing spot in *Sdk1*^CG/+ (Het) or *Sdk1*^CG/CG retinae (KO). (**C, D**) Average peak current versus expanding bright (**C**) or dark (**D**) spot diameter measured from control or knockout S1-RGCs held at –60 mV in experiments like those shown in (**A, B**). (**E, F**) Average peak current versus expanding bright (**E**) or dark (**F**) spot diameter measured from control or knockout S1-RGCs held at ~0 mV taken experiments like those shown in (**A, B**) (n = 8 for Sdk1-Het, n = 9 for Sdk1-KO). (**G**) Retinal cross-sections showing S1-RGCs in control (Het) Sdk1 knockout (KO) retinae. Scale = 50 μm. (**H**) Linescans through S1-RGC arbors in control in Sdk1 null retinae taken from experiments like those shown in (**G**). (**I**) Skeletonized S1-RGC dendrites from control (black) and Sdk1 null

*Figure 5 continued on next page*

*Figure 5 continued*

(red) retinae (scale = 50 μm). (**J–L**) Average branch number (**J**), cumulative branch length (**K**), and dendritic area (**L**) measured from control and Sdk1 null S1-RGC dendritic arbors. (**M**) Sholl analysis of dendritic arbors measured from Het and KO S1-RGCs (n = 8 for both Sdk1-Het and Sdk1-KO; *p<0.05; **p<0.01). RGC: retinal ganglion cell.

The online version of this article includes the following figure supplement(s) for figure 5:

**Figure supplement 1.** Sdk1+ Onα-RGC synaptic inputs and dendritic arbors in Sdk1 knockouts.

**Figure supplement 2.** Synaptic currents evoked by oriented bars in control and Sdk1-null S1-RGCs.

**Figure supplement 3.** Onα- and S1-RGC spatial distribution and general inner plexiform layer (IPL) lamination in Sidekick-1 (Sdk1 null retinae).

subsets targeting specific sublayers (*Figure 5—figure supplement 3*). Reconstructed ONα-RGCs in the same *Sdk1*$^{CG/CG}$ retina showed subtler deficits and more closely resembled their counterparts in heterozygote retinae (*Figure 5—figure supplement 1C–H*). Heterozygote Onα-RGCs in turn resembled wild-type Onα-RGCs labeled with the KCNG4-Cre line, supporting our functional observations, and suggesting that retinae bearing one or two copies of Sdk1 are similar (*Figure 4—figure supplement 2*). Taken together, these results show that Sdk1 loss impairs S1-RGCs dendritic arborization.

## Discussion

Here, we investigated the role of Sdk1 in retinal circuit development. In the first section of this study, we molecularly, anatomically, and functionally defined five Sdk1 interneurons (ACs and BCs) and five Sdk1-RGCs that target an inner set of IPL lamina. This family of RGCs includes two ON-DSGC types and ON-OFF S1-RGCs; the latter shows selectivity for edges located in their receptive field. In the second section, we found that Sdk1 loss caused a significant loss of S1-RGC responsivity with smaller effects on the other Sdk1-RGC types. By comparing visual responses between S1-RGCs and Onα-RGCs in control and Sdk1 nulls, we show that the loss of Sdk1 impairs S1-RGCs' responses to both bright and dark spot stimuli, as well as oriented bars. Finally, we show that these deficits arise from a selective loss of excitatory and inhibitory synaptic input on S1-RGCs and correlates with a selective loss of small branches in this neuron's dendritic arbor. We conclude that Sdk1 is required for the dendritic and synaptic development of a local edge-detecting RGC type.

### A family of Sdk1 circuits

Transcriptomic studies indicate that RGCs can be divided into at least 46 different clusters, which include several clusters that do not correspond to any known RGC and are predicted to be novel. In the process of studying Sdk1, we developed molecular signatures for three of these orphan clusters and characterized them using calcium imaging.

We show a pair of Nr2f2+ RGCs can be distinguished based on their expression of Calb and that exhibit selective responses for stimuli traveling ventrally (Nr2f2+/Calb+; C10, *Tran et al., 2019*) or dorsally (Nr2f2+/Calb-; C27, *Tran et al., 2019*) over the retina. Both cell types exhibit stronger responses to bars whose speed matches the preferred tuning of ON-DSGCs (200 μm/s) and elaborate medium-field dendritic arbors confined to sublamina 4 (*Lilley et al., 2019*; *Matsumoto et al., 2019*; *Sanes and Masland, 2015*). Moreover, reporter-labeled RGCs in the Pcdh9-Cre line, which marks ventral motion-selective ON-DSGCs (*Lilley et al., 2019*; *Matsumoto et al., 2019*), colocalize the ventral motion-selective Nr2f2+/Calb+ RGCs. Finally, we show strong RGC axon labeling in the MTN, a brain region known to be targeted by ON-DSGC axons (*Dhande et al., 2013*; *Martersteck et al., 2017*; *Oyster and Barlow, 1967*; *Yonehara et al., 2008*; *Yonehara et al., 2009*). These results strongly suggest that Nr2f2+ Sdk1 RGCs comprise two subtypes of ON-DSGCs.

S1-RGCs are a high-density, narrow dendritic field neuron, characterized by small (<200 μm) diameter dendritic arbors, strong surround suppression, and responses to both bright and dark stimuli. Similar properties have been found in several other RGCs that respond to stimuli falling in their receptive field center but are silenced when the same stimuli fall in their surround. One of these RGCs, called W3B, expresses Sdk2 and grows into sublamina 3 just like S1-RGCs, suggesting that the Sdk family plays unique roles in the development of two physically entangled, but functionally distinct local edge-detecting circuits. Four other edge-sensing types are described in a recent

atlas of functionally and anatomically defined RGC types. Of these, S1-RGCs most closely resemble type 1 or type 2 high-definition RGCs (HD1 or HD2). Like S1-RGCs, HD1 and HD2 are ON-OFF cells and can respond to edges (*Jacoby and Schwartz, 2017*). However, S1-RGCs also show sensitivity to edge orientation, which could arise from their orthogonally tuned excitatory and inhibitory inputs. This input arrangement has been observed in recently described horizontally and vertically selective OS-RGCs; however, these neurons show significantly stronger orientation selectivity compared to S1-RGCs (*Nath and Schwartz, 2016*; *Nath and Schwartz, 2017*). We cautiously speculate that S1-RGCs are a kind of local edge detector with sensitivity to edge orientation. Such a local orientation-selective RGC was reported in a recent calcium imaging survey of RGCs (G14, *Baden et al., 2016*); however, a lack of molecular markers in both studies prevented us from drawing a clear correspondence. The molecular markers and genetic access provided by our work offer a way to resolve this issue in future.

## IgSFs and circuit assembly

Several molecular recognition systems direct various steps of retinal circuit assembly, including IPL region selection (*Deans et al., 2011*; *Matsuoka et al., 2011a*; *Matsuoka et al., 2011b*; *Matsuoka et al., 2013*; *Sun et al., 2013*) and layer selection (*Duan et al., 2014*; *Duan et al., 2018*; *Ray et al., 2018*; *Yamagata and Sanes, 2008*; *Yamagata et al., 2002*). The factors that guide neurons to synapse with targets once they reach appropriate layers are the least characterized, but preliminary work in mammals and invertebrates suggests that IgSF members play a key role in this phenomenon (*Carrillo et al., 2015*; *Cosmanescu et al., 2018*; *Krishnaswamy et al., 2015*; *Peng et al., 2017*; *Sanes and Zipursky, 2020*; *Tan et al., 2015*; *Yamagata and Sanes, 2018*). Our work here adds to these findings.

We previously showed that an IgSF called Sdk2 enriches connections between VG3-ACs and W3B-RGCs, permitting this RGC to sense object motion (*Kim et al., 2015*; *Krishnaswamy et al., 2015*; *Lee et al., 2014*). In a separate study, we showed that a different IgSF, contactin 5 (Cntn5), was required for ON-OFF direction-selective RGCs to elaborate dendrites within sublamina 4 and collect input from ACs and BCs (*Peng et al., 2017*). Too few IgSFs have been studied in the context of mammalian circuit assembly to know whether the differing roles for Sdk2 and Cntn5 represent distinct roles for each IgSF or whether they represent two ends of a continuum in which IgSFs direct small-scale RGC-AC/BC interactions that lead to specific synapses. Here, we show that the closest common relative of Sdk2 and Cntn5, Sdk1, has roles in both phenomena as Sdk1 loss causes deficits in both dendritic branching and appearance of functional synapses on S1-RGCs. We hypothesize that both deficits arise because of a lack of homophilic adhesion between S1-RGCs and Sdk1+ interneurons. Our histological survey of Sdk1-interneurons suggests sublamina 3 projecting wide-field ACs, and possibly type 7 BCs, as potential interneuron targets of S1-RGCs. More characterization of these Sdk1-interneuron types, independent genetic access to these neurons, and Sdk1 misexpression methods is needed to directly test this idea.

Four RGCs express both Sdk1 and Sdk2. Effects of Sdk1 deletion on RGC types that express both Sdk1 and Sdk2 were modest (ON-DSGCs) or undetectable (ONα-RGCs). We considered the possibility that Sdk1 loss led to Sdk2 upregulation, but we did not observe a significant change in Sdk2 mRNA in Sdk1 null retinae. Simple alternatives are that Sdk1 plays a subsidiary role in these cells or that it is required for structural or functional features that we did not assay. Examining the morphology and function of Onα-RGCs and/or ON-DSGCs in Sdk2 nulls and Sdk1/Sdk2 double knockouts would be a starting point to investigate the role of Sdks in these neurons.

Exactly how Sdks exert their connectivity-enriching effects is not clear, but both proteins localize to synapses through an interaction with the MAGI family of PDZ-scaffolding proteins (*Yamagata and Sanes, 2010*). In the case of Sdk2, VG3-W3B connectivity was reduced with only minor changes to the dendritic arbors of these two neurons, suggesting that Sdks instruct synapses between interneuron-RGC pairs, which may in turn have positive effects on their dendritic arbors. Sdk1 loss causes similar losses in interneuron input to S1-RGCs but the morphological deficits on this cell are more severe, suggesting a more permissive role for Sdks in synapse formation through control of dendritic growth. The genetic and molecular definition of Sdk1 and Sdk2 expressing RGCs provided by our studies, paired with time-lapse imaging in explants, might offer a way to directly examine these alternatives.

# Materials and methods

## Key resources table

| Reagent type (species) or resource | Designation | Source or reference | Identifiers | Additional information |
|---|---|---|---|---|
| Strain, strain background (*Mus musculus*) | Sdk1-CreGFP, Sdk1$^{CG}$ | *Krishnaswamy et al., 2015* (doi: 10.1038/nature14682) | | |
| Strain, strain background (*M. musculus*) | Sdk2-CreER, Sdk1$^{CreER}$ | *Krishnaswamy et al., 2015* (doi: 10.1038/nature14682) | | |
| Strain, strain background (*M. musculus*) | Sdk1-CreER, Sdk2$^{CreER}$ | *Krishnaswamy et al., 2015* (doi: 10.1038/nature14682) | | |
| Strain, strain background (*M. musculus*) | KCNG-Cre | Jackson Laboratory | RRID:IMSR_JAX:016963 | |
| Strain, strain background (*M. musculus*) | Ai27D-ChR2-tdTomato | Jackson Laboratory | RRID:IMSR_JAX:012567 | |
| Strain, strain background (*M. musculus*) | Slc17a6-Cre | Jackson Laboratory | RRID:IMSR_JAX 016963 | |
| Strain, strain background (*M. musculus*) | Pcdh9-Cre | Mutant Mouse Research and Resource Centers | RRID:MMRRC_036084-UCD | Tissue donated by Yonehara K |
| Strain, strain background (*M. musculus*) | GCaMP6f | Jackson Laboratory | RRID:IMSR_JAX 024105 | |
| Sequence-based reagent | Sdk2-F | IDT | AssayID: Mm.PT.58.41577551 | GCTGTCCG TAAAGAAC TCCTT |
| Sequence-based reagent | Sdk2-R | IDT | AssayID: Mm.PT.58.41577551 | ATGAGGTCG TTGTACTTGGTG |
| Sequence-based reagent | Gapdh-F | *Kechad et al., 2012* (doi: 10.1523/JNEUROSCI.4127-12.2012) | Accession#:NM 008084.2 | TGCAGTGGCA AAGTGGAGAT donated by Cayouette M |
| Sequence-based reagent | Gapdh-R | *Kechad et al., 2012* (doi: 10.1523/JNEUROSCI.4127-12.2012) | Accession#:NM 008084.2 | ACTGTGCCG TTGAATTTGCC donated by Cayouette M |
| Commercial assay or kit | RNeasy Mini-kit | Qiagen | Cat#:74134 | |
| Commercial assay or kit | EZ DNAse | ThermoFisher | Cat#:11766050 | |
| Commercial assay or kit | SuperScriptIV VILO master mix | ThermoFisher | Cat#:11756050 | |
| Commercial assay or kit | PowerUp SYBR Green Master Mix | ThermoFisher | Cat#:A25741 | |
| Recombinant DNA reagent | AAVrg CAG-flex-GCaMP6f | Neurophotonics Platform Canadian Neurophotonics Platform Viral Vector Core Facility | RRID:SCR_016477 | |
| Recombinant DNA reagent | AAV9 CAG-flex-GCaMP6f | Canadian Neurophotonics Platform Viral Vector Core Facility | RRID:SCR_016477 | |
| Recombinant DNA reagent | AAV9 ef1a-flex-Tdtomato | Canadian Neurophotonics Platform Viral Vector Core Facility | RRID:SCR_016477 | |
| Recombinant DNA reagent | AAVrg-flex-Tdtomato | AddGene | Cat#:28306-AAVrg | |

*Continued on next page*

*Continued*

| Reagent type (species) or resource | Designation | Source or reference | Identifiers | Additional information |
|---|---|---|---|---|
| Antibody | Anti-DsRed (rabbit polyclonal) | Clontech Laboratories | RRID:AB_10013483 | IF(1/1000) |
| Antibody | Anti-GFP (chicken polyclonal) | Abcam | RRID:AB_300798 | IF(1/1000) |
| Antibody | Anti-Nr2f2 (mouse monoclonal) | Abcam | RRID:AB_742211 | IF(1/1000) |
| Antibody | Anti-Brn3c (mouse monoclonal) | Santa Cruz Biotechnology | RRID:AB_2167543 | IF(1/250) |
| Antibody | Anti-Nefh (mouse monoclonal) | BioLegend | RRID:AB_2314912 | IF(1/1000) |
| Antibody | Anti-calbindin | Swant | RRID:AB_2314070 | IF(1/10 000) |
| Antibody | Goat anti-osteopontin | R&D Systems | RRID:AB_2194992 | IF(1/1000) |
| Antibody | Goat anti-VAChT (goat polyclonal) | MilliporeSigma | RRID:AB_2630394 | IF(1/1000) |
| Antibody | Anti-calretinin (rabbit polyclonal) | MilliporeSigma | RRID:AB_94259 | IF(1/2000) |
| Antibody | Anti-vGlut3 (guinea pig polyclonal) | MilliporeSigma | RRID:AB_2819014 | IF(1/2000) |
| Antibody | Pig anti-RBPMS (guinea pig polyclonal) | Phosphosolutions | RRID:AB_2492226 | IF(1/100) |
| Antibody | Anti-Chx10 (goat polyclonal) | Santa Cruz Biotechnology | RRID:AB_2216006 | IF(1/300) |
| Antibody | Anti-Ap2-α (mouse monoclonal) | Developmental Studies Hybridoma Bank | Clone:3b5 | IF(1/100) |
| Antibody | Donkey anti-rabbit Alexa Fluor 405 | Abcam | RRID:AB_2715515 | IF(1/1000) |
| Antibody | Donkey anti-chicken Alexa Fluor 488 | Cedarlane | RRID:AB_2340375 | IF(1/1000) |
| Antibody | Donkey anti-goat FITC | MilliporeSigma | RRID:AB_92588 | IF(1/1000) |
| Antibody | Donkey anti-rabbit Cy3 | MilliporeSigma | RRID:AB_92588 | IF(1/1000) |
| Antibody | Donkey anti-guinea pig Cy3 | Jackson ImmunoResearch | RRID:AB_2340460 | IF(1/500) |
| Antibody | Donkey anti-goat Cy3 | MilliporeSigma | RRID:AB_92570 | IF(1/1000) |
| Antibody | Donkey anti-mouse Alexa Fluor 647 | MilliporeSigma | RRID:AB_2687879 | IF(1/1000) |
| Chemical compound, drug | Isolectin | Fisher Scientific | RRID:SCR_014365 | IF(1/200) |
| Chemical compound, drug | Sulphorhodamine 101 | MilliporeSigma | Cat#:S7635 | .2 mg/mL |
| Chemical compound, drug | Tamoxifen | MillliporeSigma | Cat#:T5648 | ~1 g/50 g body weight |
| Chemical compound, drug | Fluorescein 3000 MW dextran | Thermo Scientific | Cat#:D7156 | |
| Chemical compound, drug | QX314 Bromide | Tocris | Cat#:1014 | 5 mM |
| Chemical compound, drug | Ames Medium | MilliporeSigma | Cat#:A1420 | |
| Software, algorithm | ImageJ | Fiji | https://imagej.net/Fiji | |
| Software, algorithm | MATLAB | Simulink | https://www.mathworks.com/products/matlab.html | |
| Software, algorithm | Simple Neurite Tracer | https://imagej.net/plugins/snt/ | RRID:SCR_016566 | |
| Software, algorithm | Trees Toolbox | https://www.treestoolbox.org/ | RRID:SCR_010457 | |

*Continued on next page*

*Continued*

| Reagent type (species) or resource | Designation | Source or reference | Identifiers | Additional information |
|---|---|---|---|---|
| Software, algorithm | RStudio | RStudio | https://rstudio.com | |
| Software, algorithm | ggplot2 | https://ggplot2.tidyverse.org/ | RRID:SCR_014601 | |

## Animals

Animals were used in accordance with the rules and regulations established by the Canadian Council on Animal Care and protocols were approved by the Animal Care Committee at McGill University. Male and female Sdk1-CreGFP (Sdk1[CG]), Sdk1-CreER (Sdk1[CreER]), and Sdk2-CreER mice aged 35–100 days old were used in this study. Details about the generation of these lines can be found in previous studies (*Krishnaswamy et al., 2015*; *Yamagata and Sanes, 2018*). Rosa26-LSL-ChR2-tdTomato mice were obtained from the Jackson Laboratory (AID27, Jackson Labs, RRID:IMSR_JAX:012567) and crossed with Sdk1CreER mice for some anatomical experiments. *Slc17a6*-Cre mice were obtained from the Jackson Laboratory (Jackson Labs, RRID:IMSR_JAX 016963) and crossed with Cre-dependent GCaMP6f lines (AI95D, Jackson Labs, RRID:IMSR_JAX 024105). KCNG4-Cre mice were obtained from the Jackson Laboratory (KCNG4-Cre, Jackson Labs, RRID: IMSR_JAX: 029414) and crossed with Cre-dependent tdtomato lines (AID27).

## Viruses

AAVrg CAG-flex-GCaMP6f, AAV9 CAG-flex-GCaMP6f, and AAV9 ef1a-flex-Tdtomato viral vectors were purchased from the Canadian Neurophotonics Platform Viral Vector Core Facility (RRID:SCR_016477) and AAVrg-flex-Tdtomato was purchased from Addgene (viral prep: 28306-AAVrg). Retrogradely infecting AAVs were used for brain injections while AAV9 was used for intravitreal injections.

## Injections

AAVs were injected either intraocularly or intracranially to label Sdk1-RGCs. For intracranial injections, mice were anesthetized using isoflurane (2.5% in $O_2$), given a combination of subcutaneous carprofen and local bupivacaine/lidocaine mix for analgesia, transferred to a stereotaxic apparatus, and a small craniotomy (<1 mm) made in the appropriate location on the skull using a dental drill. Next, a Neuros syringe (65460–03, Hamilton, Reno, NV) filled with virus was lowered into either the LGN (2.15 mm posterior from bregma, 2.27 mm lateral from the midline and 2.75 mm below the pia) or SC (3.85 mm posterior from bregma, 0.75 mm lateral from the midline and 1.5 mm below the pia) using a stereotaxic manipulator. A microsyringe pump (UMP3-4, World Precision Instrument, Sarasota, FL) was used to infuse 400 nL of virus (15 nL/s) bilaterally in dLGN or SC and the bolus allowed to equilibrate for 8 min before removing the needle. For intraocular injections, mice were anesthetized as above, given carprofen as analgesic, and 1 µL of virus injected via an incision posterior to the eye's ora serrata using a bevelled Hamilton syringe (7803-05, 7634-01, Hamilton). Mice were given at least 2 weeks to recover before experimental use.

## Tamoxifen

Tamoxifen (TMX) (MillliporeSigma, Cat#T5648) was dissolved in anhydrous ethanol at 200 mg/mL, diluted in sunflower oil to 10 mg/mL, sonicated at 40°C until dissolved, and stored at –20 °C. Prior to injection, TMX aliquots were heated to 37° C and delivered intraperitoneally at ~1 g/50 g body weight to Sdk1[CreER]xAi27D mice. The dose was repeated twice over 2 days and mice were used between 2 and 4 weeks following treatment.

## Histology

Mice were euthanized by isoflurane overdose and transcardially perfused with chilled PBS followed by 4% (w/v) paraformaldehyde (PFA) in PBS and enucleated. Eye cups were fixed for an additional 45 min and brains fixed for an additional 2–5 hr in chilled 4% (w/v) PFA. For whole-mount stains, tissue was incubated in primary antibodies for 7 days at 4 °C and incubated in secondary antibodies overnight at 4 °C following a series of washes. Stains following two-photon calcium imaging were performed similarly but included lectin to stain blood vessels for registration. Identical procedures were used to stain

thick brain sections obtained cutting 100 -µm-thick slices from tissue embedded in 2.5% low EEO agarose (MilliporeSigma, A0576) using a compresstome (VF-200-0Z, Precisionary Instruments, Greenville, NC). For cross-sections, post-fixed retinae were sunk in 40% (w/v) sucrose/PBS, transferred to embedding agent (OCT, Tissue-Plus, Fischer Scientific), flash frozen in 2-methylbutane at –45 °C, and then sectioned onto slides at 30 µm thickness on a cryostat. For immunostaining, slides were first washed in PBS, blocked with blocking solution (4% normal donkey serum, 0.4% Triton-X-100 in PBS) for 2 hr and incubated overnight with primary antibodies at 4 °C. Tissue was then washed with PBS and incubated in secondary antibodies for 2 hr at room temperature prior to the final wash and tissue mounting.

## Antibodies and blood vessel stains

Antibodies used were as follows: rabbit anti-DsRed (1:1000, Clontech Laboratories; RRID:AB_10013483); chicken anti-GFP (1:1000, Abcam, Cambridge, UK; RRID:AB_300798); mouse anti-Nr2f2 (1:1000, Abcam; RRID:AB_742211); mouse anti-Brn3c (1:250, Santa Cruz Biotechnology, Dallas, TX; RRID:AB_2167543); mouse anti-Nefh (SMI-32, 1:1000, BioLegend, San Diego, CA; RRID:AB_2314912); rabbit anti-calbindin (1:10,000, Swant, Marly, CH; RRID:AB_2314070); goat anti-osteopontin (1:1000, R&D Systems, Minneapolis, MN; RRID:AB_2194992); goat anti-VAChT (1:1000, MilliporeSigma, Darmstadt, DE; RRID:AB_2630394); rabbit anti-calretinin (1:2000, MilliporeSigma; RRID:AB_94259); guinea pig anti-vGlut3 (1:2000, MilliporeSigma; RRID:AB_2819014); guinea pig anti-RBPMS (1:100, Phosphosolutions, Aurora, CO; RRID:AB_2492226); mouse anti-Chx10 (1:300, Santa Cruz Biotechnology; RRID:AB_2216006); and mouse anti-Ap2-α (1:100, clone 3b5 from Developmental Studies Hybridoma Bank, Iowa City, IA). Secondary antibodies were conjugated to Alexa Fluor 405 (Abcam; RRID: AB_2715515), Alexa Fluor 488 (Cedarlane, Ontario, CA; RRID:AB_2340375), FITC (MilliporeSigma; RRID:AB_92588), Cy3 (MilliporeSigma; RRID:AB_92588, RRID:AB_92570, or Jackson ImmunoResearch, West Grove, PA; RRID:AB_2340460) or Alexa Fluor 647 (MilliporeSigma; RRID:AB_2687879). Isolectin (Fisher Scientific, Waltham, MA; RRID:SCR_014365) was incubated along with the secondary antibodies when applicable.

## Confocal imaging and analysis

Images of stained tissue were acquired on an Olympus FV1000 confocal laser-scanning microscope or a Zeiss LSM-710 inverted confocal microscope (Advanced BioImaging Facility, McGill University) at a resolution of 512 × 512 or 1024 × 1024 pixels with step sizes ranging from 0.45 µm to 8 µm. Following image acquisition, minor processing of images including landmark correspondence transforms to match two-photon and stained retinal fields as well as stitching of large tilescans was performed using ImageJ.

For density recovery profile analysis (DRP), RGC soma centers were selected manually on ImageJ and their x–y coordinates used to generate DRPs using code modeled after *Rodieck, 1991*. Briefly, intercellular distances from each cell to every cell were computed using Pythagoras' theorem and binned into 15-µm-wide annuli centered upon the reference cell; for Onα- and M2-RGCs, annulus width equaled 25 µm. Counts within each annulus were divided by annulus area, averaged across a given condition, fitted with a sigmoid curve for visual clarity, and plotted alongside DRPs obtained from the entire GFP+ population for comparison. The density of each Sdk1-RGC type was obtained by dividing marker-labeled cells by image area. These values were averaged for each type and compared to mean density of all GFP+ cells as a bar graph. The mean cell density obtained from this analysis was used to assess the proportion of cells labeled by Sdk1 among BCs, ACs, and RGCs.

We analyzed dendritic morphology as described previously (*Krishnaswamy et al., 2015*). Briefly, images of single S1-RGCs and Onα RGCs taken from P30-60 retinae were traced manually through the z-stack using simple neurite tracer (ImageJ). Path ROIs describing neuronal processes were converted to stacks and analyzed for morphological properties using the Trees Toolbox on MATLAB (*Cuntz et al., 2010*). The *len_tree*, *B_tree* and *vhull* functions in the Trees Toolbox were used to measure the cumulative length of all dendritic branches, branch length, and arbor area, respectively.

To obtain mean IPL projection depth of each Sdk1+ type, we took linescans of pixel intensity across the IPL from images stained with antibodies to VAChT (sublamina 2 and 4) or reporter, normalized these signals to the maximum intensity, and averaged these traces across each condition. IPL depth is expressed as a percent and sublaminae judged by the position of the peak intensities in the VAChT

channel. We applied the straighten transform (ImageJ) to correct the VAChT bands on a few curved sections and applied the same transform to reporter channels prior to linescan procedure.

## Calcium imaging

Calcium imaging was performed as previously described (*Liu et al., 2018*). Briefly, mice were dark adapted for at least 2 hr, euthanized, and then retinae rapidly dissected under infrared illumination into oxygenated (95% $O_2$; 5% $CO_2$) Ames solution (MilliporeSigma, A1420). Next, retinae were mounted onto a filter paper with the RGC layer facing up, placed in a recording chamber, mounted on the stage of a custom-built two-photon microscope, and perfused with oxygenated Ames solution warmed to 32–34°C. Responses of GCaMP6*f+* RGCs to visual stimuli delivered through the objective were imaged at 920 nm and collected at 8 Hz. Each image plane (360 × 72 µm) of the movie contained GCaMP fluorescence, SR101 fluorescence, stage coordinates, and visual stimulus synchronization pulses to permit offline analysis. A few microliters of sulphorhodamine 101 (SR101, 2 mg/mL, MilliporeSigma, Cat#S7635) was added to the recording chamber to label blood vessels and a map of the main blood vessels emanating from the optic disk acquired for post hoc image registration.

## Post hoc image registration and RGC ROI selection

Following each calcium imaging session, averages of each calcium imaged field were stitched into a single image using custom software written in MATLAB (*Rochon et al., 2021*; Dryad: https://doi.org/10.5061/dryad.4xgxd2593). These large images contained a map of the blood vessel pattern surrounding the retinal optic disk and imaged fields (*Figure 2—figure supplement 1A and B*). These images were then scaled into microns, overlayed on images of post hoc stained retinae, and rotated or translated until the blood vessel patterns between the two datasets aligned (*Figure 2—figure supplement 1A–C*). Fine blood vessel morphology was then used as inputs to the landmark correspondences function in Fiji to perform final alignment between these two fields (*Figure 2—figure supplement 1D–K*). ROIs of stained neurons were used to extract responses from two-photon movies (*Figure 4—figure supplement 1L–O*). Osteopontin distribution, which is enriched in the dorsal-temporal quadrant of the retina (*Bleckert et al., 2014*), permitted alignment of RGC calcium responses to the cardinal axes. For a subset of our studies, we were able to restain retinae with two additional markers, register these restains as above, and together with the trace similarity score (see below) categorize Sdk1-RGCs.

## Calcium imaging analysis

RGC responses corresponding to stimulus synchronization pulses were extracted, aligned, and analyzed using custom software written in MATLAB (Simulink). Responses to stimuli were expressed as a z-score where the mean and standard deviation were obtained from an equal duration of the prestimulus baseline on each trial. We computed three higher descriptive statistics: the ON-OFF index was calculated as described previously (*Baden et al., 2016*), but briefly

$$On - Off\,Index = \frac{ON_{resp} - Off_{resp}}{ON_{resp} + Off_{resp}}$$

where $ON_{resp}$ and $OFF_{resp}$ is the mean cell response over a bright and dark full-field stimulus, respectively. ON-OFF indexes from a dataset in which all RGCs express GCaMP6f (Slc17a6-Cre::AI27) were plotted alongside Sdk1-RGC ON-OFF indexes after filtering this dataset using a quality index computed as described in *Baden et al., 2016*. A direction selectivity index was calculated using the circular variance of the cell response for all eight moving bar directions (*Nath and Schwartz, 2016*):

$$DSI = \frac{|\sum\left(Resp_\theta \cdot e^{i\theta}\right)|}{\sum\left(Resp_\theta\right)}$$

where $Resp_\theta$ is the maximum cell response for each bar direction. A similar approach was done to compute an orientation selectivity index, where instead of the first complex exponential the second ($e^{2i\theta}$) is used. A similarity score was used to measure each functional RGC group's similarity to the group mean. This score is the cosine between the group mean and each individual trace in our dataset. Both traces were Euclidean normalized prior to this measurement using the *norm* function in MATLAB.

## Electrophysiology

Retinae for electrophysiological recordings were prepared and visualized as described for calcium imaging. For cell-attached recordings, the patch electrodes (4–5 MΩ) were filled with Ames solution. For whole-cell recordings, patch electrodes (5–7 MΩ) were filled with an internal solution containing 112 mM Cs methanosulfate, 10 mM NaAc, 0.2 mM $CaCl_2$, 1 mM $MgCl_2$, 10 mM EGTA, 8 mM CsCl, and 10 mM HEPES (pH 7.4). In both cell-attached and whole-cell recordings, fluorescein 3000 MW dextran (Thermo Scientific, D7156) was added to make the electrode visible under two-photon illumination. For whole-cell recordings, internal solution was supplemented with 5 mM QX314 Bromide. Excitatory currents and inhibitory currents were isolated by adjusting the holding potential to match reversal potentials for excitation (0 mV) and inhibition ($E_{Cl}$ ~ –60 mV). Signals from loose-patch and whole-cell recordings were acquired with a MultiClamp 700B amplifier (Molecular Devices) and digitized at 20 kHz using custom software written in LabView. For spikes, the MultiClamp was put into I = 0 mode and Bessel filter set at 1 kHz. For currents, the MultiClamp was put in VC mode and Bessel filter set at 3 kHz. Analysis of electrophysiological signals was performed in MATLAB (Simulink) as follows. Briefly, action potentials were detected in loose patch recordings using the *peakfinder* function and binned (50 ms) over the entire length of the trial; firing rate histograms for each trial were then averaged and subjected to further processing based on each stimulus. Direction and orientation-selective indices were computed from mean firing rate histograms as described for calcium imaging data. For whole-cell currents, trials were averaged, peak amplitude measured, and integral were computed using the *trapz* function across each stimulus epoch. For stationary bar stimuli, integrated currents for each bar flash were normalized to average maximum in controls, plotted against bar orientation for each cell, and then peak responses aligned and averaged across all RGCs in heterozygote or knockout retinae.

## Visual stimuli

A DLP light crafter (Texas Instruments, Dallas, TX) was used to project monochrome (410 nm) visual stimuli through a custom lens assembly that steered stimulus patterns into the back of a 20× objective (*Euler et al., 2009*). All visual stimuli were written in MATLAB using the psychophysics toolbox and displayed with a background intensity set to $1 \times 10^4$ R*/rod/s. Custom electronics were made to synchronize the projector LED to the scan retrace of the two-photon microscope.

For calcium imaging experiments, visual stimuli were centered around the microscope field of view. Moving bar stimuli consisted of a bright bar moving along its long axis in one of eight directions. The bar was 1200 μm wide, 3200 μm long moving at either 960 μm/s (fast) or 240 μm/s (slow). For electro-physiological experiments, the cell-receptive field center was identified using a grid of flashing spots and a small user-controlled probe and the location with the highest response assigned as the center for all subsequent stimuli. Moving bars were thinned for electrophysiological experiments to 200 μm; length and speed were the same as calcium imaging studies. Expanding spot stimuli began as a small circle that flashed at the receptive field center and grew after each ON-OFF flash. Stationary, rotating bars had the same dimensions as those for the moving bar but flashed and rotated through nine orientations. All stimuli were preceded by a gray background whose duration equaled the stimulus duration in each trial.

## RNAseq data analysis

RNA sequencing data was taken from the publicly available Broad Institute Single Cell Portal for three previous studies (*Shekhar et al., 2016*; *Tran et al., 2019*; *Yan et al., 2020*). To identify type-specific marker for Sdk1-RGCs we first compared expression of markers between these neurons and all other RGCs to find a differentially expressed subset. Next, we analyzed this subset for the presence five-gene combinations that could segregate each Sdk1 type; Nr2f2 was identified from *Rheaume et al., 2018*. Sequencing data plots were generated with the ggplot plot package in R.

## qPCR

For RT qPCR, standard procedures were followed (*Kechad et al., 2012*). Briefly, RNA was extracted from retinas of both Sdk1-Het and Sdk1-KO mice using the RNEasy Mini-kit (Qiagen, Hilden, Germany, Cat#74134) and cDNA synthesized using EZ DNAse (ThermoFisher, Cat#11766050) and SuperScriptTMIV VILO master mix (ThermoFisher, Cat#11656050). Quantitative PCR was done using the PowerUp SYBR Green Master Mix (ThermoFisher, Cat#A25741) on a ViiA 7 Real-Time PCR System.

Primers used were as follows: Sdk2-F: GCTGTCCGTAAAGAACTCCTT; Sdk2-R: ATGAGGTCGTTG TACTTGGTG; GAPDH-F: TGCAGTGGCAAAGTGGAGAT; GAPDH-R: ACTGTGCCGTTGAATTTGCC. The Sdk2 levels were normalized to the abundance of the housekeeping gene Gapdh to obtain a relative expression level.

## Statistics

No statistical method was used to predetermine sample size. Statistical comparisons between Sdk1-knockout and Sdk1-heterozygote electrophysiological, morphological, and calcium imaging data were performed using the *anova1* function in MATLAB (Simulink) followed by the *multcompare* function for pairwise testing.

## Acknowledgements

We thank K Yonehara for gifting us fixed retinal tissue from PCDH9-Cre mice and M Cayouette and C Jolicoeur for advice, help, and reagents for Sdk2 RT qPCR studies. We also thank Drs J Sanes, E Cooper, S Trenholm, K Yonehara, E Cook, and E Feinberg for helpful comments and suggestions on our manuscript.

## Additional information

### Funding

| Funder | Grant reference number | Author |
| --- | --- | --- |
| Canadian Institutes of Health Research | project grant | Arjun Krishnaswamy |
| Natural Sciences and Engineering Research Council of Canada | discovery grant | Arjun Krishnaswamy |
| Canadian Institutes of Health Research | Graduate Fellowship | Pierre-Luc Rochon |
| Fonds de Recherche du Québec - Santé | Graduate Fellowship | Aline Giselle Rangel Olguin |
| Consejo Nacional de Ciencia y Tecnología | Graduate Fellowship | Aline Giselle Rangel Olguin |

The funders had no role in study design, data collection and interpretation, or the decision to submit the work for publication.

### Author contributions

Pierre-Luc Rochon, Conceptualization, Formal analysis, Investigation, Methodology, Visualization, Writing - original draft, Writing - review and editing; Catherine Theriault, Investigation, Methodology, Writing - review and editing; Aline Giselle Rangel Olguin, Formal analysis, Investigation, Methodology, Writing - review and editing; Arjun Krishnaswamy, Conceptualization, Funding acquisition, Investigation, Methodology, Project administration, Supervision, Visualization, Writing - original draft, Writing - review and editing

### Author ORCIDs

Arjun Krishnaswamy (iD) http://orcid.org/0000-0002-7706-4657

### Ethics

Animals were used in accordance with the rules and regulations established by the Canadian Council on Animal Care and protocol (2017-7889) was approved by the Animal Care Committee at McGill University.

### Decision letter and Author response

Decision letter https://doi.org/10.7554/eLife.70870.sa1
Author response https://doi.org/10.7554/eLife.70870.sa2

## Additional files

### Supplementary files
• Transparent reporting form

### Data availability
Sample registration fields, registration code, and GCaMP6f datasets are available at Dryad (https://doi.org/10.5061/dryad.4xgxd2593).

The following dataset was generated:

| Author(s) | Year | Dataset title | Dataset URL | Database and Identifier |
|---|---|---|---|---|
| RochonP-L TC, Rangel Olguin AG, Krishnaswamy A | 2021 | Data From: The cell adhesion molecule Sdk1 shapes assembly of a retinal circuit that detects localized edges | https://doi.org/10.5061/dryad.4xgxd2593 | Dryad Digital Repository, 10.5061/dryad.4xgxd2593 |

The following previously published datasets were used:

| Author(s) | Year | Dataset title | Dataset URL | Database and Identifier |
|---|---|---|---|---|
| Benhar I, Hong G, Yan W, Adiconis X, Arnold ME, Lee JM, Levin JZ, Lin D, Wang C, Lieber CM, Regev A, He Z, Sanes JR, Tran NM, Shekhar K, Whitney IE | 2019 | Study: Mouse retinal ganglion cell adult atlas and optic nerve crush time series | https://singlecell.broadinstitute.org/single_cell/study/SCP509/mouse-retinal-ganglion-cell-adult-atlas-and-optic-nerve-crush-time-series?genes=Sdk1&cluster=Crush%20RGCs&spatialGroups=--&annotation=Cluster--group--cluster&subsample=1000&tab=distribution#study-summary | Broad Institute Single Cell Portal, Cluster--group--cluster |
| Yan W, Laboulaye MA, Tran NM, Whitney IE, Benhar I, Sanes JR | 2020 | Study: Mouse Retinal Cell Atlas: Molecular Identification of over Sixty Amacrine Cell Types | https://singlecell.broadinstitute.org/single_cell/study/SCP919/mouse-retinal-cell-atlas-molecular-identification-of-over-sixty-amacrine-cell-types | Broad Institute Single Cell Portal, SCP919 |
| Shekhar K, Lapan SW, Whitney IE, Tran NM, Macosko EZ, Kowalczyk M, Adiconis X, Levin JZ, Nemesh J, Goldman M, McCarroll SA, Cepko CL, Regev A, Sanes JR | 2016 | Study: Retinal Bipolar Neuron Drop-seq | https://singlecell.broadinstitute.org/single_cell/study/SCP3/retinal-bipolar-neuron-drop-seq | Broad Institute Single Cell Portal, SCP3 |

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
