## [Decision Letter]

**Acceptance summary:**

This study represents a significance contribution to our understanding of molecular factors which govern development of neural circuits. The breadth of techniques used is extremely impressive and the authors are to be commended for such a thorough, interesting and important study.

**Decision letter after peer review:**

Thank you for submitting your article "The cell adhesion molecule Sdk1 shapes assembly of a retinal circuit that detects localized edges" for consideration by *eLife*. Your article has been reviewed by 3 peer reviewers, one of whom is a member of our Board of Reviewing Editors and the evaluation has been overseen by Gary Westbrook as the Senior Editor. The reviewers have opted to remain anonymous.

The reviewers have discussed their reviews with one another, and the Reviewing Editor has drafted this to help you prepare a revised submission. The full reviews are attached at the end. The summary of the Essential Revisions should be used to guide your revision.

Essential Revisions:

(1) Include some additional data to compare the heterzygot Sdk1+/- to WT. Reviewer #2, is quite specific about what would be acceptable.

(2) Further characterization of S1- RGCs would strengthen the finding in that it could confirm whether they are indeed edge detecting cells.

(3) If possible, it would be great to determine whether there is an upregulation of other Sdks in RGCs subtypes that are not impacted in Sdk1 ko. Along the same lines, all reviewers have asked for a larger discussion Sdks and related proteins and retinal wiring.

(4) All reviewers have commented on mosaic analysis – their issues should be addressed.

*Reviewer #1 (Recommendations for the authors):*

A major goal of developmental neurobiology is to understand the molecular cues that are involved in the wiring up of neural circuits. In this study, the research is carried out in the retina where there are several features that make it a strong candidate to address these questions-namely cell types are well defined and the locations of synapses between different cell types are clearly demarcated in plexiform layers within which axons and dendrites of the neurons that comprise distinct microcircuits co-stratify.

The authors take advantage of this system to explore the role that a candidate molecular marker from immunoglobulin superfamily of adhesion molecules called Sidekicks (Sdks). A prior study showed that Sdk2 was critical for matching of a particular interneuron with a particular subtype of RGC. Here the authors explore Sdk1.

The authors use a large array of anatomical markers (derived from lists of genes in recent RNA transcript libraries) to identify the 5-6 different RGC subtypes that express Sdk1. They use multiple methods to confirm this assignment to cell types including morphology, immunofluorescence, projection patterns to the brain and light responses. They go on to show that in the Sdk1-ko mouse, only one of these subtypes exhibits a large phenotype – a significant reduction in the light responses. Using whole cell voltage clamp recordings, they show that his reduction in light response is correlated with a is a reduction in excitatory and inhibitory synaptic inputs, consistent with a lack of synapse formation. This is a convincing set of data that they have identified a factor that is critical for the formation of a microcircuit within the retina. There is also a bigger picture, which is that this class of Sdk2 adhesion molecules may be part of a molecular code used by multiple circuits to wire up during development.

This is an extremely thorough study that identifies the role that Sidekick 1 – a member of the IgSF family of adhesion molecules – plays in the wiring up a particular microcircuits in the retina. First the authors use multiple methods to identify the cells that express Sdk1 – identifying 5-6 different RGC subtypes and a few amacrine and bipolar cell subtypes. Using information from RGC sequencing atlases, they identify candidate proteins that can be used to further classify these RGC subtypes and are able to positively identify 5 of them – 2 that were named subtypes and 3 that were not. Taking advantage of some genetic tricks (i.e. sparse labeling of SDK-1-Cre x with flex mouse, non-sparse labeling for anterograde projections, retrograde AAVs injected in L and SC) the authors were able to use immunofluorescence, morphology and projections patterns to further classify the cells. Finally, they characterize the light responses if these cells using calcium imaging. By combining all of these methods, they make a strong case that they have labeled ON-α RGCs, M2-iRGCs two different ON direction selective GCs and a difficult-to-classify cell – -the S1. Note the S1 cells send dendrites in S3 of the IPL, which the authors show also contains processes from excitatory and inhibitory interneurons that are also Sdk1 positive. Hence these are good candidate for Sdk1-dependent circuit-wiring.

They then look in the Sdk1 KO mouse. First they use 2p calcium imaging to assess the impact on all the different Sdk1+ RGCs. They identify the subtypes post imaging by finding the same field of view and using antibody staining (though this method has now been used by a few labs, they do an excellent job describing their method in a supplemental figure). They find deficits in actually quite a few cell types but this biggest impact is on the S1-RGCs, which have a greatly reduced light response.

Using the intersection between GFP expression under SDK1-Cre-GFP fusion mouse + retrograde labeling from SC/LGN, they are able to target electrodes to just two subtypes of Sdk1+ RGCs, the S1s and the On-alphas. Hence they can directly compare the impact of the knockout on spiking properties and synaptic inputs to these two cell types. They see dramatic reduction in light responses and EPSCs and IPSCs onto these cells in S1-RGCs. In contrast, they see no phenotype at all on On-α RGCs. This is strong evidence that there is reduction in synapse formation onto S1 cells but not all Sdk1+ RGC subtypes.

In some ways the impact of this study seems small – one adhesion molecule impacts one microcircuit within the retina. But I would argue that the methods used and the completeness of the study represent a broader impact. There were many important insights – for example multiple cells might have a particular adhesion molecule but it is required for only one microcircuit. Also, the authors have identified more markers for different RGC types and have done this classification in the context of development. These sorts of observations are critical for ultimately understanding the molecular code for wiring up the retina. For these reasons, I think this study represent a significant step forward in the field.

I do think in a revision that perhaps the authors can zoom out a bit and give the readers a bigger picture. How big is the Sdk family of proteins? Are they alternatively spliced? How numerous are they compared to say protocadherins, or semaphorins or other factors that have been implicated. Again – I think this will make it more relevant for broader audience.

Below are a few comments that will hopefully clarify the presentation.

1. The one major issue I have is with Figure 1 H-K. The authors are strongly recommended to read a recent review (Keely, Elgen and Reese,JCN, 2020) arguing that density recover profiles that do not take into account soma size are not revealing mosaic organization and therefore this really cannot be used to "verify" they have a cell type. That said. I don't think the lack of the mosaic analysis will not significantly weaken their results -- the antibody staining + physiology is sufficient for indicating that they have different cell types.

2. (Lines 108-114/Figure 1 T-U )- the authors want to argue that Ost+/Nefh- are M2-ipRGCs and so they look at projections to the MTN via anterograde labeling. However, it is unclear how they correlate anterograde projections with the Ost+/Nefh- RGCs in retina. This needs to be clarified.

3. Labels in Figure 1are very hard to read. Also, I strongly recommend that the authors refer to the different RGC subtypes in subsequent figures as their better known subtypes (M2, α, ON-DSGC) rather than the immunipositive proteins. It was quite confusing. – I had to print out Figure 1V and use it several times to figure out what was going on in several figures.

4. For Figure 2 – it was not clear when the authors were using GCaMP in all RGCs (expressed under vglut-Cre – line 163) or specific for Sdk1-Cre (line 138).

5. Figure 2 – is bar velocity 1000mm/sec (which I believe is the speed of an action potential!) or 1000 µm/sec?

6. Figure 2 – would be great if there were some quantification go figures H and I – are all Pcdh-Cre stained for Nr2f2? What percentage for Calb+?

7. Figure 3 – there seems like a substantial loss of DS in the ON-DSGCs that the authors should note. Again, this broadens the impact of the study because clearly Sdk1 is important for this circuit, but the impact is more subtle. It is not that they need to study it here but it is important to emphasis the multiple impacts it has. Hence line is 220-221 are not quite accurate.

*Reviewer #2 (Recommendations for the authors):*

In this study Rochon et al., elucidate the distribution of an adhesion molecule called Sidekick 1 (Sdk1) among retinal neurons. Sidekicks are a family of cell adhesion molecules, implicated in organization of dendritic arbors among retinal neurons, and also in regulating the organization of synapses across specific visual circuits. The authors show that Sdk1 is selectively expressed among 5 retinal output (ganglion cell) types, 3 inhibitory interneuron (amacrine) cells and 2 excitatory interneuron (bipolar) cell types, all in the inner retina. Using a combination of genetic tools and cell specific markers, the authors detail the identity of the cell types expressing Sdk1 in the retina. Using a loss of function approach, the authors show that lack of Sdk1 impacts the synaptic connectivity and visual response properties of a specific retinal output neuron (retinal ganglion cell or RGC) type that responds to local edges. This interesting work reveals a selective role of Sdk1 for the function of a specific visual circuit, and as such the authors succeeded in the task they set out to achieve i.e., determine the distribution and an insight into possible functional role(s) of Sdk1 in the retina. The strengths of this study are the combinatorial approach of genetic tools, electrophysiology and functional imaging and the data are of a high standard.

There are some weaknesses though, in the current version of the manuscript that if addressed would strengthen the work and its implications.

(i) The current version used Sdk1 HET animals (with one copy of Sdk1) as controls for all functional and morphological experiments, but it is unclear how cells in the HET retina compare in terms of visual responses and morphology to cells in a wildtype retina with two copies of Sdk1. It is thus difficult to extrapolate the magnitude of visual response deficits or dendritic mis-organization of cells in the Sdk1 deficient retinas.

(ii) The feature of the S1-RGC responding to local edges could be detailed further to include consideration of potential temporal sensitivity/tuning of the cells and asymmetry between their ON and OFF response profiles as is often observed for other RGCs that respond to local edges. Comparison between the ON vs OFF components in control retina are also of merit since the OFF responses are impacted more than the ON responses in the Sdk1 deficient retinas, underscoring a potential asymmetry in how Sdk1 could wire these RGCs.

(iii) The authors show that Sdk1 could be implicated in regulating the synaptic connectivity of a specific retinal ganglion cell (S1-RGC) type that responds to local edges but do not discuss whether these interactions could be homophilic or heterophilic and whether or not the 5 interneurons that express Sdk1 could serve as presynaptic partners to the S1-RGC (or the other RGC types expressing Sdk1). Such a discussion would be of benefit in consideration of the fact that some RGC types that express Sdk1, like the ON DS or ONalpha, have known presynaptic bipolar cell and amacrine cell partners that lack Sdk1 expression. Additionally, most of the amacrine and bipolar cell types shown to express Sdk1 do not co-laminate with the S1-RGC. Thus, a heterophilic mode of interaction for Sdk1's wiring of specific visual circuits seems to be likely which appears different from the homophilic interactions that underlie Sdk2 connections between VG3 amacrine cells and W3B RGCs that both express Sdk2 (authors' previous work). Because Sdk1 is a close relative of Sdk2 that appears to operate differently, a discussion would enable readers to better grasp the context and significance of the work.

(iv) Even though Sdk1 is expressed by 5 different ganglion cell types, its loss seems to selectively perturb the synaptic connectivity and functional responses of one specific (S1-RGC) ganglion type, indicating that perhaps other members of the Sdk family could compensate for lack of Sdk1 in the unaffected ganglion cell types. Inclusion of data indicating whether or not Sdk2 or other Sdk1 relatives are upregulated in these unaffected ganglion cell types in Sdk1 deficient retinas, would clarify the redundancy between these IgSF members during retinal circuit formation and also clarify why the loss of Sdk1 leaves certain retinal circuits unperturbed.

The following concerns should be addressed by the authors:

(1) How close is the Sdk1 HET retina to a wildtype retina with two copies of Sdk1? Could the authors compare RGC responses they obtain in the Sdk1 HET to responses in a wildtype retina to enable the reader to grasp the magnitude of deficits upon Sdk1 deficiency? As the authors have done an incredible job of findings cell specific markers for each of the RGC types that express Sdk1, they could use these markers to correlate responses in HET vs wildtype retina. At the minimum, responses from easily identifiable RGC types like the ONalpha or the ON DS which the authors have a specific transgenic line for, could be recorded in retinas with two copies of Sdk1 to evaluate how similar these are compared to the responses (using the same stimuli and lighting conditions) obtained from HET retinas with a single Sdk1 copy. This data would be helpful to validate the use of the Sdk1 HET as a control.

(2) Given that previous accounts of edge sensitive RGC types like the HD RGCs show that these cells are most responsive to specific speeds (Jacoby and Schwartz 2017), it would be worth determining if the S1-RGC type detailed in this work are also tuned to a specific speed of stimuli. This aspect of temporal tuning would aid the characterization of the response profile of the S1-RGC type. The authors would most likely already have this data as they have made a similar temporal tuning determination for the ON DS RGCs.

(3) Along the same lines, edge sensing RGCs can have ON and OFF responses that are not symmetrical (Jacoby and Schwartz 2017; Roska and Werblin 2001). As such it would add to the characterization of the S1-RGC response profile, if the authors could determine if the ON and OFF responses of the S1-RGC are symmetric/equivalent or not. From the examples shown in Figure 4 it appears as though the ON responses are more prominent than the OFF both for the expanding spot and orientation experiments i.e., more spikes allocated during light increments. Other edge sensing RGCs can have one component (ON or OFF) more dominant and thus it would be important for the reader to know whether the S1-RGC shows a similar asymmetry or not. Again, the authors already have this data so it should be straight forward to analyze and perhaps include plots where the ON and OFF responses (# of spikes) of the S1-RGCs are compared in the same plot.

(4) The authors indicate that Sdk2 might be compensating for Sdk1 in the RGC types unaffected by Sdk1 deficiency. However, they do not provide evidence for whether or not Sdk2 is upregulated in the unaffected RGCs upon Sdk1 deficiency. The authors have generated antibodies to Sdk2 in their previous work which could be used to determine whether or not Sdk2 expression is upregulated amongst select RGC types upon Sdk1 deficiency (this could be compared in HET vs knockout).

(5) The authors bring up the homophilic vs heterophilic interactions between IgSFs like Sdks in the Introduction section when discussing the role of IgSFs like Sdks for synapse formation but do not include a discussion of whether Sdk1 could be operating via homophilic or heterophilic interactions to set up synaptic connections in the inner retina. Given the Sdk1 expression profiles it appears that Sdk1 could operate via heterophilic interactions, which is different from how Sdk2 wires the VG3 amacrine cell-W3B RGC circuit and it would be of benefit to the reader to include a discussion of the mechanistic basis of Sdk1 interactions in the inner retina and speculation of which Sdk1 presynaptic partners (amacrine or bipolar) are likely to be engaged or not.

(6) The authors need to include more details as to how the mean cell density analysis was done from the density recovery profiles as the current version of the manuscript lacks sufficient detail for this analysis.

*Reviewer #3 (Recommendations for the authors):*

This paper evaluates the role of sidekick-1, a recognition molecule that is part of the immunoglobulin superfamily. Sidekick-1 is expressed in interneurons and 3 previously-defined ganglion cell types, including the ON α, 2 ON-DS, and a novel edge detector S1-RGC. The elimination of the sidekick 1 has the most severe effects on the S1-RGC, causing its dendritic arbors to be less extensive, to receive fewer excitatory and inhibitory inputs, and in losing its orientation tuning. The knockout of sidekick-1 has less severe effects on the other ganglion cell types. This work fits into the great challenge of identifying cell types by molecular, functional, and morphological parameters. The authors have focused on one type of ganglion cell that expresses sidekick-1 and use calcium imaging, single cell recordings, and morphological assessment to demonstrate what this cell might do functionally and how it is perturbed in the absence of sidekick-1. The strength of this paper is the demonstration that elimination of Sidekick-1 has effects on the inputs to these ganglion cells. Perhaps the main weaknesses of this paper are (1) the S1 ganglion cell's functional classification as weakly tuned orientation selective and an edge detector has not been shown convincingly, and (2) how and why these cells are changed in the absence of Sidekick-1 remains speculative, yet the mechanism is attributed to sidekick-1's role in synapse formation, which has not been demonstrated directly.

Specific comments:

The authors show that high levels of Sdk2 in On-α, M2, and ON-DSGCs compensate for the loss of Sdk1 and attenuates phenotypes. To elucidate whether intact phenotypes are truly due to compensatory increase in Sdk2 expression or the functional properties of these 3 RGC types are independent of Sdk1, it would be helpful to understand the relative expression levels of Sdk1 and Sdk2 in Sdk1-expressing RGCs. Without such data, it is difficult to distinguish between lack of a role for Sdk-1 vs a compensatory effect of Sdk2 expression in the other cell types (α, ON DS).

The authors call these ganglion cells local edge detectors without clear evidence in Figure 2.

This system used by the authors has the advantage of gaining mechanistic insight into how this recognition molecule works in the development of a well-defined circuit, yet the authors fall short of providing this insight.

The authors should confirm if there are any morphological and physiological changes in Sdk1 heterozygotes compared to control retina. (Note from Reviewing editor: experiments proposed by Reviewer 2 would address this).

Figure 1: If possible, the authors also show the ganglion cell axon terminals in the LGN and SC labeled from the Cre expression in the eye. This is important to establish that the AAV-based retrograde labeling in Figure 4 is working as the authors think it is.

Figure 1 H-K, Figure 2H: The multiple channels are difficult to evaluate. The authors should show in the supplementary material the immunolabeling in individual channels.

It is unclear what about Figure 2E demonstrates that S1-RGCs are responding to local edges since there is not information about receptive field sizes yet. Figure 2 shows that the S1-RGCs have weak OS, are not DS, and have ON-OFF light responses. Only the information from Figure 4 reveals that these cells may be local edge detectors. (Note from Reviewing editor: experiments proposed by Reviewer 2 would address this).

Figure 3. Sholl analysis is not showing a striking difference between the control and Sdk-1 KO. The ON α ganglion cells also have differences at 3 locations with significance. The authors' claim of the relative stability of ON α is not compelling and should soften these claims.

The authors speculate that the orthogonal tunings of excitatory and inhibitory inputs contribute to orientation-selective properties of S1-RGCs. While excitatory inputs retain orientation-tuning properties, inhibitory inputs completely lose the orientation selectivity. To elucidate how the loss of Sdk1 affects inhibitory inputs from amacrine cells, it would be helpful to investigate the morphology of 3 kinds of Sdk1-ACs that the author discovered in KO mice. How does the loss of inhibitory orientation-selectivity affect spike output? As shown in the supplementary material, the most striking effect of the Sdk1 KO is the obliteration of inhibition to orientation selectivity. Can the authors better elucidate the contributions of each of these inputs to orientation selectivity? Clarification of these issues is needed.

The authors demonstrate that the two mostly likely partners of the S1 RGC is the wide field amacrine cell and the type 7 ON cone bipolar cell. Have the authors examined the interaction of these interneurons with the S1 RGC in the Sidekick-1 KO? Additional data would make this compelling but if that does not exist, the authors should soften these claims.

If the major change in the Sidekick-1 KO is a change in the dendritic field sizes of S1 RGCs, then the question remains whether the mosaic of these cells is disrupted in the KO through lack of homophilic homotypic interactions. The authors may want to soften these claims.

Although the authors state that the overall structure of IPL remain unchanged in KO retina, it would be helpful to see the quantification of depth of the entire retina or IPL to see if the loss of Sdk1 affects retinal structures.

Supplementary Figure 5 lacks quantification and demonstrates some differences in RBPMS, VGlut3, Ap2-α staining.

Supplementary Figure 5 lacks a diagram of lamination.

The authors posit that "Sdk1 promotes synapse formation between these neurons and their interneuron targets." To elucidate the roles of Sdk1 in synapse formation, it requires morphological investigation of excitatory and inhibitory synapses onto S1 ganglion cells or physiological investigation of mEPSCs and mIPSCs in control and KO retina. Without such data, it would be difficult to conclude that the role of Sdk1 is in synapse formation.

Lines 322-328. The authors land on synapse formation has the deficit in the Sidekick-1 KO; however, this is not apparent from the data. If the authors would like to make the main point about synapse formation, then quantifying synaptic puncta would be necessary. The authors could consider either changing their claim or providing additional data.

---

## [Author Response]

Essential revisions:(1) Include some additional data to compare the heterzygot Sdk1+/- to WT. Reviewer #2, is quite specific about what would be acceptable.

We now provide new data on Sdk1+ Onα-RGCs in the KCNG4-Cre line which shows that the anatomy and visual responses of these wild-type RGCs are like their counterparts in Sdk1 heterozygotes.

(2) Further characterization of S1- RGCs would strengthen the finding in that it could confirm whether they are indeed edge detecting cells.

As requested, we now provide new analysis of S1-RGC ON and OFF responses with detail about their symmetry and speed preference. Briefly, S1-RGC ON-OFF balance varies with stimulus size, with stimuli larger than their receptive field center producing stronger ON versus OFF responses. S1-RGCs also prefer bars moving at ~200µm/sec to those moving at ~1000µm/sec.

(3) If possible, it would be great to determine whether there is an upregulation of other Sdks in RGCs subtypes that are not impacted in Sdk1 ko. Along the same lines, all reviewers have asked for a larger discussion Sdks and related proteins and retinal wiring.

We now provide measurements of Sdk2 transcript levels in whole retinae taken from Sdk1-heterozygotes and Sdk1 knockouts. At this resolution, Sdk1 loss does not result in statistically significant increase in Sdk2 mRNA. We have incorporated this result with a broader text on Sdks and retinal wiring in the discussion.

(4) All reviewers have commented on mosaic analysis – their issues should be addressed.

We now provide extra detail regarding our analysis of the spatial distribution of Sdk1 RGCs, have modified our description of this method to reflect the limitations described in Keeley et al., 2020, and have included new mosaic analysis in Sdk1-KOs as requested by reviewer #3.

Reviewer #1 (Recommendations for the authors):Below are a few comments that will hopefully clarify the presentation.1. The one major issue I have is with Figure 1 H-K. The authors are strongly recommended to read a recent review (Keely, Elgen and Reese,JCN, 2020) arguing that density recover profiles that do not take into account soma size are not revealing mosaic organization and therefore this really cannot be used to "verify" they have a cell type. That said. I don't think the lack of the mosaic analysis will not significantly weaken their results -- the antibody staining + physiology is sufficient for indicating that they have different cell types.

We thank the reviewer for recommending this recent paper on the limitations of the density recovery profile (DRP) in assessment of cell type. We did not intend to use the DRP as our sole assessment of cell-type and agree with the reviewer that the combination of approaches we used (histology, anatomy, physiology) are sufficient to show different RGC types. We have now softened our language regarding the use of this measurement and have referenced this review. We have also clarified our figures to indicate that the grey lines show the density of all GFP labelled RGCs in the Sdk1-CreGFP line.

2. (Lines 108-114/Figure 1 T-U )- the authors want to argue that Ost+/Nefh- are M2-ipRGCs and so they look at projections to the MTN via anterograde labeling. However, it is unclear how they correlate anterograde projections with the Ost+/Nefh- RGCs in retina. This needs to be clarified.

We appreciate that Reviewer 1 has thought carefully about our labelling studies. In short, we cannot directly assign staining from the MTN to specific Sdk1 RGCs in the retina using anterograde labelling. Our goal here was to illustrate why retrograde labelling from the lateral geniculate nucleus (LGN) and superior colliculus (SC) preferentially labelled the Onα− and S1-RGCs. The presence of Sdk1-RGC axon labeling in MTN and OPN are consistent with the idea that Sdk1 labels non-image forming types that might project more strongly to these areas than they do to LGN/SC. We have clarified this portion of the text.

3. Labels in Figure 1are very hard to read. Also, I strongly recommend that the authors refer to the different RGC subtypes in subsequent figures as their better known subtypes (M2, α, ON-DSGC) rather than the immunipositive proteins. It was quite confusing. – I had to print out Figure 1V and use it several times to figure out what was going on in several figures.

We apologize to the reviewer for the difficult figure labels and use of immunopositive proteins to refer to Sdk1 RGCs rather than their subtype nomenclature. We have increased the size of the figure labels and have replaced immunopostive protein labels with RGC type wherever possible.

4. For Figure 2 – it was not clear when the authors were using GCaMP in all RGCs (expressed under vglut-Cre – line 163) or specific for Sdk1-Cre (line 138).

We apologize for our lack of clarity in this figure. We wanted to give readers a way to compare the ON-OFF index for each Sdk1-RGC group to the full RGC population. The grey lines in the On-Off index graphs shown in Figures 2B-E were obtained by calcium imaging GCaMP6f-expressing RGCs in retinae taken from the Vglut2-Cre/Rosa-LSL-GCaMP6f line. All other RGC data in this figure comes from Sdk1-RGCs infected with AAVs bearing Cre-dependent GCaMP6f. We have further clarified this in the text and the legends.

5. Figure 2 – is bar velocity 1000mm/sec (which I believe is the speed of an action potential!) or 1000 µm/sec?

We apologize for the careless mistake. The reviewer is correct the bars moved at 1000µm/sec.

6. Figure 2 – would be great if there were some quantification go figures H and I – are all Pcdh-Cre stained for Nr2f2? What percentage for Calb+?

We obtained a few retinae as a gift from Dr. K. Yonehara and stained them with Nr2f2 and Calb. The Pcdh9-Cre retinae we analyzed show ~1035cells/mm2 in the GCL and we now include the proportions of Nr2f2+/Calb+/Pcdh9-Cre+ cells in the text.

7. Figure 3 – there seems like a substantial loss of DS in the ON-DSGCs that the authors should note. Again, this broadens the impact of the study because clearly Sdk1 is important for this circuit, but the impact is more subtle. It is not that they need to study it here but it is important to emphasis the multiple impacts it has. Hence line is 220-221 are not quite accurate.

This is a good point. In an effort to be cautious about our interpretation we may have inadvertently downplayed phenotypes on Sdk1+ ON-DSGC types. We have altered lines 220-221 to incorporate this point.

Reviewer #2 (Recommendations for the authors):The following concerns should be addressed by the authors:(1) How close is the Sdk1 HET retina to a wildtype retina with two copies of Sdk1? Could the authors compare RGC responses they obtain in the Sdk1 HET to responses in a wildtype retina to enable the reader to grasp the magnitude of deficits upon Sdk1 deficiency? As the authors have done an incredible job of findings cell specific markers for each of the RGC types that express Sdk1, they could use these markers to correlate responses in HET vs wildtype retina. At the minimum, responses from easily identifiable RGC types like the ONalpha or the ON DS which the authors have a specific transgenic line for, could be recorded in retinas with two copies of Sdk1 to evaluate how similar these are compared to the responses (using the same stimuli and lighting conditions) obtained from HET retinas with a single Sdk1 copy. This data would be helpful to validate the use of the Sdk1 HET as a control.

The reviewer asks whether Sdk1-RGCs in heterozygotes show an intermediate phenotype relative to the same RGCs bearing two functional copies of Sdk1. We acknowledge the reviewer’s concern and provide recordings and dendritic reconstructions from ONα-RGCs genetically targeted using the KCNG4-Cre mouse line. These data are shown in Figure 4 —figure supplement 2 and show that the morphology and physiology of Onα-RGCs bearing two copies or one copy of Sdk1 are similar.

(2) Given that previous accounts of edge sensitive RGC types like the HD RGCs show that these cells are most responsive to specific speeds (Jacoby and Schwartz 2017), it would be worth determining if the S1-RGC type detailed in this work are also tuned to a specific speed of stimuli. This aspect of temporal tuning would aid the characterization of the response profile of the S1-RGC type. The authors would most likely already have this data as they have made a similar temporal tuning determination for the ON DS RGCs.

This is a good point. We have now provided a comparison of speed tuning for the S1-RGC and ONα-RGC obtained from our calcium imaging experiments. These results show that S1-RGC responses are stronger for bars moving at 200µm/sec versus bars moving at 1000µm/sec.

(3) Along the same lines, edge sensing RGCs can have ON and OFF responses that are not symmetrical (Jacoby and Schwartz 2017; Roska and Werblin 2001). As such it would add to the characterization of the S1-RGC response profile, if the authors could determine if the ON and OFF responses of the S1-RGC are symmetric/equivalent or not. From the examples shown in Figure 4 it appears as though the ON responses are more prominent than the OFF both for the expanding spot and orientation experiments i.e., more spikes allocated during light increments. Other edge sensing RGCs can have one component (ON or OFF) more dominant and thus it would be important for the reader to know whether the S1-RGC shows a similar asymmetry or not. Again, the authors already have this data so it should be straight forward to analyze and perhaps include plots where the ON and OFF responses (# of spikes) of the S1-RGCs are compared in the same plot.

We appreciate that Reviewer 2 has thought deeply about the response properties of S1-RGCs in relation to other edge-sensing RGC types. We now provide example raster plots of S1-RGC firing to a centered spot and full field flash, as well as graphs comparing their ON and OFF firing rate versus spot diameter. These data are now presented as Figure 4 —figure supplement 1 and show that S1-RGCs are ON-dominated if stimuli are larger than their receptive field center.

(4) The authors indicate that Sdk2 might be compensating for Sdk1 in the RGC types unaffected by Sdk1 deficiency. However, they do not provide evidence for whether or not Sdk2 is upregulated in the unaffected RGCs upon Sdk1 deficiency. The authors have generated antibodies to Sdk2 in their previous work which could be used to determine whether or not Sdk2 expression is upregulated amongst select RGC types upon Sdk1 deficiency (this could be compared in HET vs knockout).

This is a great idea, but difficult for a couple of reasons. First, the Sdk2 antibody stain appears as small puncta with no detectable signal in somata. Second, Sdk1/2 double positive and Sdk2-exclusive RGCs, such as W3B-RGCs, all ramify in the same region of the IPL. It would be hard to assign these puncta to Sdk1/Sdk2 double-positive RGCs and compare their intensity/number between controls and Sdk1 nulls. Instead, we have performed whole-retina real-time qPCR for Sdk2 from Sdk1-Hets and Sdk1-KOs. These results are shown in Figure 3 —figure supplement 1C and no significant elevation of Sdk2 in the absence of Sdk1. We have incorporated this result into our results and discussion on Sdks and IgSFs.

(5) The authors bring up the homophilic vs heterophilic interactions between IgSFs like Sdks in the Introduction section when discussing the role of IgSFs like Sdks for synapse formation but do not include a discussion of whether Sdk1 could be operating via homophilic or heterophilic interactions to set up synaptic connections in the inner retina. Given the Sdk1 expression profiles it appears that Sdk1 could operate via heterophilic interactions, which is different from how Sdk2 wires the VG3 amacrine cell-W3B RGC circuit and it would be of benefit to the reader to include a discussion of the mechanistic basis of Sdk1 interactions in the inner retina and speculation of which Sdk1 presynaptic partners (amacrine or bipolar) are likely to be engaged or not.

We have now expanded our discussion of the differences between Sdk1 and Sdk2 which includes some discussion about potential synaptic partners.

(6) The authors need to include more details as to how the mean cell density analysis was done from the density recovery profiles as the current version of the manuscript lacks sufficient detail for this analysis.

We apologize for the lack of detail. We have now provided a more thorough discussion of how mean cell density analysis was performed in the methods and clarified their presentation in the manuscript.

Reviewer #3 (Recommendations for the authors):The authors show that high levels of Sdk2 in On-α, M2, and ON-DSGCs compensate for the loss of Sdk1 and attenuates phenotypes. To elucidate whether intact phenotypes are truly due to compensatory increase in Sdk2 expression or the functional properties of these 3 RGC types are independent of Sdk1, it would be helpful to understand the relative expression levels of Sdk1 and Sdk2 in Sdk1-expressing RGCs. Without such data, it is difficult to distinguish between lack of a role for Sdk-1 vs a compensatory effect of Sdk2 expression in the other cell types (α, ON DS).The authors call these ganglion cells local edge detectors without clear evidence in Figure 2.

We have altered the title of this segment to reflect the data shown up until Figure 2. It now reads:

“Brn3c+ RGCs are ON-OFF RGCs that respond to bars travelling along the same axis”

This system used by the authors has the advantage of gaining mechanistic insight into how this recognition molecule works in the development of a well-defined circuit, yet the authors fall short of providing this insight.The authors should confirm if there are any morphological and physiological changes in Sdk1 heterozygotes compared to control retina. (Note from Reviewing editor: experiments proposed by Reviewer 2 would address this).

The reviewer asks whether Sdk1 heterozygotes show an intermediate phenotype relative to animals bearing two functional copies of Sdk1. We acknowledge the reviewer’s concern and provide recordings and dendritic reconstructions from ONα RGCs genetically targeted using the KCNG4-Cre mouse line. These data are shown in Figure 4 —figure supplement 2 and show that the morphology and physiology of Onα-RGCs bearing two copies or one copy of Sdk1 are similar.

Figure 1: If possible, the authors also show the ganglion cell axon terminals in the LGN and SC labeled from the Cre expression in the eye. This is important to establish that the AAV-based retrograde labeling in Figure 4 is working as the authors think it is.

We agree with the reviewer that LGN/SC sections showing Sdk1-RGC terminals would reinforce our method to retrogradely label Sdk1 RGCs. We have now grouped this data along with our other brain projection data in Figure 1 —figure supplement 3.

Figure 1 H-K, Figure 2H: The multiple channels are difficult to evaluate. The authors should show in the supplementary material the immunolabeling in individual channels.

We apologize for the difficult presentation of this data. We now show the immunolabelling in individual channels in a new supplemental figure (Figure 1 —figure supplement 2).

Agreed. This sentence was an undetected fragment from an earlier draft of this manuscript. Please see our response to point#5.

Figure 3. Sholl analysis is not showing a striking difference between the control and Sdk-1 KO. The ON α ganglion cells also have differences at 3 locations with significance. The authors' claim of the relative stability of ON α is not compelling and should soften these claims.

It is a good point. In our effort to be cautious about interpreting the effects of Sdk1 loss we may have inadvertently downplayed potential effects on the Onα-RGC. We have now softened our claims about the absence of phenotypes from the ONα-RGC and included new discussion of potential functional phenotypes on the ON-DSGCs in the Sdk1-KO as recommended by reviewer 1.

The authors speculate that the orthogonal tunings of excitatory and inhibitory inputs contribute to orientation-selective properties of S1-RGCs. While excitatory inputs retain orientation-tuning properties, inhibitory inputs completely lose the orientation selectivity. To elucidate how the loss of Sdk1 affects inhibitory inputs from amacrine cells, it would be helpful to investigate the morphology of 3 kinds of Sdk1-ACs that the author discovered in KO mice. How does the loss of inhibitory orientation-selectivity affect spike output? As shown in the supplementary material, the most striking effect of the Sdk1 KO is the obliteration of inhibition to orientation selectivity. Can the authors better elucidate the contributions of each of these inputs to orientation selectivity? Clarification of these issues is needed.

The reviewer wants to know how loss of inhibitory input from Sdk1-ACs causes the visual response deficits seen in S1-RGCs. This is a great idea, but a direct test is methodologically not possible since we cannot perturb the Sdk1-ACs independently of the Sdk1-RGCs and assess the consequences on S1-RGC function. We are in the process of generating intersectional reagents to specifically label the S1-RGC and Sdk1 interneurons to better characterize their visual responses and potential synaptic interactions. These experiments are still in early stages and are months away from completion. We now bring up this important idea in our revised discussion.

The authors demonstrate that the two mostly likely partners of the S1 RGC is the wide field amacrine cell and the type 7 ON cone bipolar cell. Have the authors examined the interaction of these interneurons with the S1 RGC in the Sidekick-1 KO? Additional data would make this compelling but if that does not exist, the authors should soften these claims.

We too are excited by the prospect that the likely partners of S1-RGCs would be the sublamina-3 targeting wide-field amacrine and type 7 bipolar cells. In short, we have not examined their interaction because we currently cannot access these neurons without also affecting the Sdk1-RGCs. An intersectional approach might be feasible but will require months of development and troubleshooting to deploy. We have softened these claims.

If the major change in the Sidekick-1 KO is a change in the dendritic field sizes of S1 RGCs, then the question remains whether the mosaic of these cells is disrupted in the KO through lack of homophilic homotypic interactions. The authors may want to soften these claims.

It is a good point, we now include mosaic analysis of Onα- and S1-RGCs in Sdk1-KOs and compare them to those found in Sdk1-Hets in Figure 5 —figure supplement 3. Density recovery profiles of both RGCs in Sdk1 nulls resemble those found in controls.

Although the authors state that the overall structure of IPL remain unchanged in KO retina, it would be helpful to see the quantification of depth of the entire retina or IPL to see if the loss of Sdk1 affects retinal structures.Supplementary Figure 5 lacks quantification and demonstrates some differences in RBPMS, VGlut3, Ap2-α staining.

We apologize for this careless omission and poor choice of representative crossections. We have updated the panels in this figure showing that gross laminar morphology in Sdk1 nulls resembles that found in controls.

Supplementary Figure 5 lacks a diagram of lamination.

We apologize for this careless omission and indicate IPL and nuclear layer landmarks for these panels.

The authors posit that "Sdk1 promotes synapse formation between these neurons and their interneuron targets." To elucidate the roles of Sdk1 in synapse formation, it requires morphological investigation of excitatory and inhibitory synapses onto S1 ganglion cells or physiological investigation of mEPSCs and mIPSCs in control and KO retina. Without such data, it would be difficult to conclude that the role of Sdk1 is in synapse formation.

We agree. Our data show a deficit in functional synaptic input on S1-RGCs and a structural correlate in the form of reduced dendritic arborization. Sdk1 could be involved in promoting or stabilizing synapses, dendrites, or both. We have removed this phrase from the results and have better balanced these alternatives in the discussion.

Lines 322-328. The authors land on synapse formation has the deficit in the Sidekick-1 KO; however, this is not apparent from the data. If the authors would like to make the main point about synapse formation, then quantifying synaptic puncta would be necessary. The authors could consider either changing their claim or providing additional data.